# Amyloid-associated increases in soluble tau relate to tau aggregation rates and cognitive decline in early Alzheimer's disease

Alexa Pichet Binette ⬤[1,92] ✉, Nicolai Franzmeier ⬤[2,92], Nicola Spotorno[1], Michael Ewers ⬤[2,3], Matthias Brendel ⬤[4,5], Davina Biel[2], Alzheimer's Disease Neuroimaging Initiative*, Olof Strandberg[1], Shorena Janelidze[1], Sebastian Palmqvist ⬤[1,6], Niklas Mattsson-Carlgren[1,7,8], Ruben Smith ⬤[1,7], Erik Stomrud[1,6], Rik Ossenkoppele ⬤[1,9] & Oskar Hansson ⬤[1,6] ✉

For optimal design of anti-amyloid-β (Aβ) and anti-tau clinical trials, we need to better understand the pathophysiological cascade of Aβ- and tau-related processes. Therefore, we set out to investigate how Aβ and soluble phosphorylated tau (p-tau) relate to the accumulation of tau aggregates assessed with PET and subsequent cognitive decline across the Alzheimer's disease (AD) continuum. Using human cross-sectional and longitudinal neuroimaging and cognitive assessment data, we show that in early stages of AD, increased concentration of soluble CSF p-tau is strongly associated with accumulation of insoluble tau aggregates across the brain, and CSF p-tau levels mediate the effect of Aβ on tau aggregation. Further, higher soluble p-tau concentrations are mainly related to faster accumulation of tau aggregates in the regions with strong functional connectivity to individual tau epicenters. In this early stage, higher soluble p-tau concentrations is associated with cognitive decline, which is mediated by faster increase of tau aggregates. In contrast, in AD dementia, when Aβ fibrils and soluble p-tau levels have plateaued, cognitive decline is related to the accumulation rate of insoluble tau aggregates. Our data suggest that therapeutic approaches reducing soluble p-tau levels might be most favorable in early AD, before widespread insoluble tau aggregates.

Alzheimer's disease (AD) is characterized by a cascade of molecular and neurodegenerative brain changes, in which Aβ plaques start to accumulate ~20 years before symptom onset followed by the accumulation and spreading of neurofibrillary tau aggregates with ensuing neurodegeneration and AD dementia[1,2]. Multiple neuroimaging studies using positron emission tomography (PET) to track in vivo pathological processes in AD showed that cortical Aβ deposition precedes neocortical tau aggregation by several years[1,3], and Aβ has been shown to amplify tau spreading in preclinical studies[4,5]. Among the earliest tau-related abnormalities in AD are increases in soluble

[1]Clinical Memory Research Unit, Faculty of Medicine, Lund University, Lund 205 02, Sweden. [2]Institute for Stroke and Dementia Research (ISD), University Hospital, LMU Munich, Munich, Germany. [3]German Center for Neurodegenerative Diseases (DZNE), Munich, Germany. [4]Department of Nuclear Medicine, University Hospital, LMU Munich, Munich, Germany. [5]Munich Cluster for Systems Neurology (SyNergy), Munich, Germany. [6]Memory Clinic, Skåne University Hospital, Malmö, Sweden. [7]Department of Neurology, Skåne University Hospital, Lund 205 02, Sweden. [8]Wallenberg Center for Molecular Medicine, Lund University, Lund, Sweden. [9]Alzheimer Center Amsterdam, Department of Neurology, Amsterdam Neuroscience, Vrije Universiteit Amsterdam, Amsterdam UMC, Amsterdam, The Netherlands. [92]These authors contributed equally: Alexa Pichet Binette, Nicolai Franzmeier. *A list of authors and their affiliations appears at the end of the paper. ✉e-mail: alexa.pichet_binette@med.lu.se; oskar.hansson@med.lu.se

## Table 1 | BioFINDER-2 cohort demographics

|  | Aβ-negative controls (n = 204) | Aβ-positive non-demented (n = 130) | Aβ-positive AD dementia (n = 66) |
|---|---|---|---|
| Age (years) | 62.4 ± 15.0 | 71.7 ± 8.4 | 72.7 ± 7.3 |
| Sex F (%F) | 100 (49%) | 63 (49%) | 37 (56%) |
| Education (years) | 12.7 ± 3.2 | 12.4 ± 4.4 | 11.8 ± 4.3 |
| APOEε4 carriers (%) | 73 (36%) | 95 (73%) | 44 (67%) |
| CSF p-tau217 (pg/ml) | 47.6 ± 31.4 | 243.7 ± 174.5 | 570.8 ± 300.9 |
| MMSE | 29.0 ± 1.2 | 27.7 ± 1.9 | 20.1 ± 4.4 |
| Cognitive composite score | 0.01 ± 0.7 | −1.4 ± 1.2 | −4.4 ± 1.5 |
| tau-PET follow-up time (years) | 2.0 ± 0.6 | 2.0 ± 0.6 | 1.7 ± 0.3 |
| Number of PET scans, n (%) |  |  |  |
| 2 | 173 | 92 | 48 |
| 3 | 29 | 32 | 18 |
| 4 | 2 | 6 | 0 |
| Cognitive follow-up time (years) | 2.4 ± 0.8 | 3.1 ± 0.7 | 2.0 ± 0.6 |
| Number of cognitive assessments, n (%) |  |  |  |
| 2 | 144 | 13 | 11 |
| 3 | 42 | 52 | 22 |
| 4 | 12 | 46 | 33 |
| 5 | 4 | 19 | 0 |

Data are presented as mean ± standard deviation unless specified otherwise. One Aβ-positive non-demented participant had missing APOE genotype. Cognitive composite are z-scores. See Supplementary Table 1 for ADNI participants.
Aβ beta-amyloid, APOEε4 apolipoprotein E genotype (carrying at least one ε4 allele), CSF p-tau217 cerebrospinal fluid phosphorylated tau 217, MMSE Mini-Mental State Examination, PET positron emission tomography.

hyperphosphorylated tau (p-tau) concentrations, which can reliably be detected in cerebrospinal fluid (CSF)[1] and precede the neocortical deposition of insoluble fibrillary tau pathology in AD[6–8]. The rate of cellular tau production[9] and soluble p-tau increases[10] have been shown to correlate with Aβ burden, further strengthening the view that Aβ systematically induces p-tau increases. Recent studies could show that levels of p-tau start increasing already at the preclinical stage of AD when persons are asymptomatic[11], increase even further in early symptomatic AD and reach a plateau in patients with AD dementia[12,13]. Importantly, levels of soluble p-tau have been shown to correlate with neuropathological levels of insoluble fibrillar tau[14–16], and preclinical studies found that soluble hyperphosphorylated p-tau seeds can induce a cascade of tau aggregation in animal models[4,17–19]. Therefore, increases in soluble p-tau may drive subsequent tau aggregation and spread in AD, suggesting p-tau as a potential treatment target to halt tau spread and the formation of toxic insoluble tau aggregates. However, the cross-link between Aβ, p-tau and tau aggregation is yet to be better understood in humans across the spectrum of AD.

A putative mechanism by which Aβ triggers tau secretion is via stimulation of neuronal activity[20]. Extracellular Aβ has been consistently shown to induce neuronal hyperactivity[21,22] and neuronal activity enhances tau secretion[17,23,24]. Therefore, a major pathway for the propagation of tau pathology in AD are synaptic connections, which are assumed to provide the roadmap for trans-neuronal tau spreading[25,26]. This is supported by preclinical work, showing that injected p-tau seeds can be taken-up by post-synaptic neurons and spread across anatomically connected rather than spatially adjacent brain regions[4,17–19], further supporting soluble p-tau as a key driver of tau propagation. We recently translated these preclinical findings to in vivo neuroimaging data by combining functional MRI-based connectomics with tau-PET in AD. We and others reported that patient-level tau accumulation is related to the connectivity patterns of regions where tau aggregates emerge first (i.e, tau epicenters)[27–29].

Altogether, the literature supports a view in which i) Aβ is linked with increased concentrations of soluble p-tau early in the AD continuum, ii) greater concentrations of soluble p-tau are associated with insoluble tau aggregates, and iii) tau aggregates expand across connected brain regions. Yet, evidence in human studies overwhelmingly stem from cross-sectional studies, and a biologically plausible cascade model that incorporates these individual findings and can link soluble p-tau with downstream tau aggregation and clinical manifestations across the entire AD spectrum is still missing. Testing this model is critical to clarify the potential of p-tau as a target to attenuate tau aggregation and spread, and the optimal target populations that may benefit from such treatments. This is particularly important since drugs efficiently lowering the soluble tau levels have recently advanced to phase 2 (Clinical Trial ID: NCT05399888). Based on progress in the anti-tau drug development pipeline, it is essential to establish during which phases of the disease soluble p-tau levels are most associated with subsequent accumulation of insoluble tau aggregates and cognitive decline.

To address these knowledge gaps, we leveraged both cross-sectional Aβ and CSF p-tau levels, resting-state fMRI connectomics, longitudinal tau-PET (measuring accumulation of insoluble tau aggregates) and longitudinal cognitive assessments (measuring cognitive decline) in two independent large-scale AD cohort studies. We first tested how soluble p-tau concentrations measured in CSF and local Aβ fibrils at baseline affected the local accumulation rate of tau aggregates over time across the brain. Second, adding resting-state functional connectivity, we investigated if soluble p-tau was related to connectivity-based accumulation of tau aggregates across the brain. Lastly, we investigated those biomarker changes leading to cognitive decline, bringing together both measures of soluble p-tau and tau aggregates accumulation as they follow more closely the apparition of symptoms than Aβ pathology[30,31]. Informed by the results we propose a working model of pathophysiological AD progression that is largely built on longitudinal in vivo data, from the early preclinical phase towards the clinical syndrome of AD dementia. Overall, our study suggests that Aβ-related increased soluble levels of p-tau is a key driver in the accumulation of tau aggregates and connectivity-mediated tau spreading in early-stage preclinical and prodromal AD, while the accumulation rate of tau aggregates becomes self-promoting in late-stage AD dementia and might no longer be driven by p-tau. We thus suggest that soluble p-tau levels may be an optimal treatment target for attenuating tau aggregation and subsequent cognitive decline in the earliest stages of AD, prior to the development of widespread insoluble fibrillar tau aggregates and dementia, which will be critical for the design of future clinical trials.

## Results

### Study design and participants

The study was conducted in one of the largest samples available worldwide with cross-sectional Aβ-PET and soluble CSF p-tau measurement, and longitudinal tau-PET and cognition, covering the full spectrum of AD. The Swedish BioFINDER-2 cohort was the primary cohort of interest, from which participants with all abovementioned measures were included. The final sample included 400 participants composed of 204 cognitively unimpaired (CU) Aβ-negative participants, 65 Aβ-positive CU, 65 Aβ-positive patients with mild cognitive impairment (MCI) and 66 Aβ-positive individuals with AD dementia (Table 1). The CU and MCI Aβ-positive participants were grouped together to study the early stages of AD, hereafter referred to as "non-demented participants". All main analyses were also conducted in each group separately, with all results in Supplementary information. To quantify the regional levels of insoluble tau aggregates, tau-PET was performed using the second-generation radiotracer [18F]RO948,

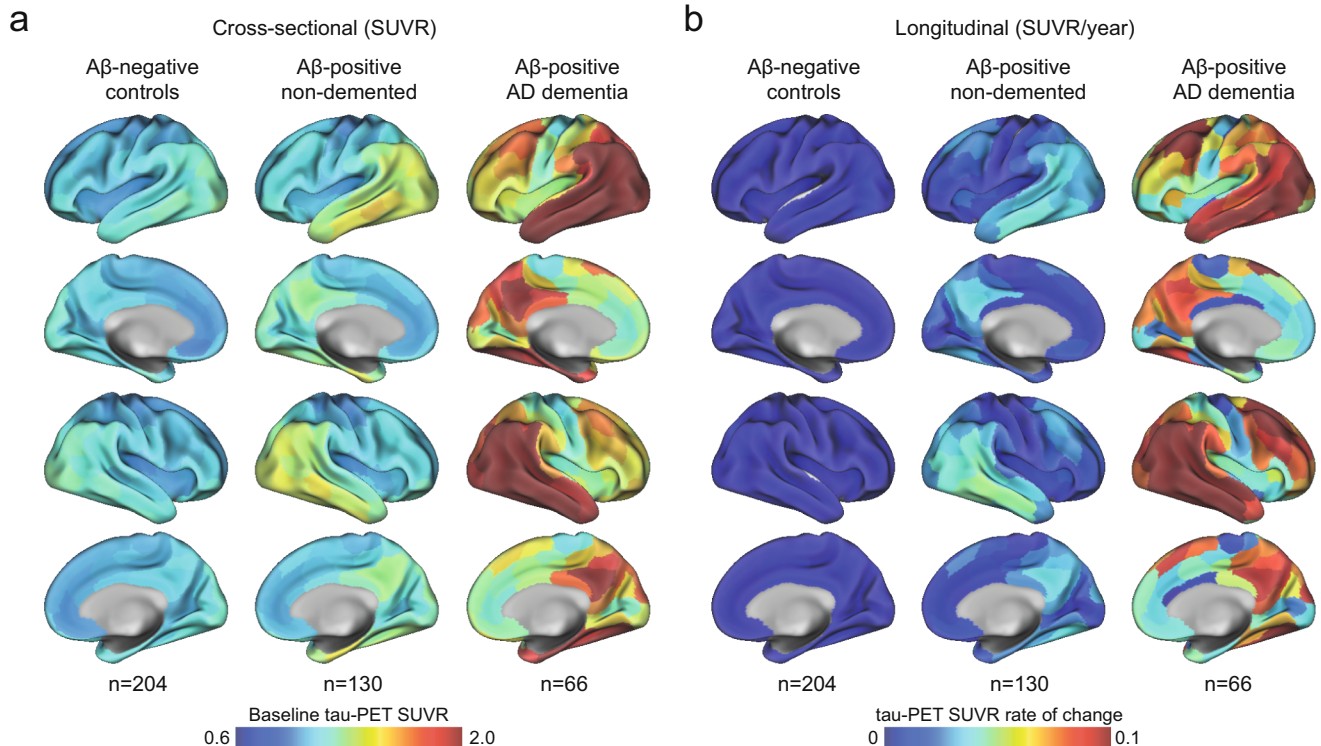

**Fig. 1 | Mean spatial distribution of cross-sectional tau-PET [$^{18}$F]RO948 SUVR and longitudinal rate of change. a** Surface renderings of average baseline tau-PET SUVR in Aβ-negative controls, Aβ-positive non-demented participants and Aβ-positive patients with AD dementia in the 200 parcels from the Schaefer 200-ROI atlas. **b** Surface renderings of yearly tau-PET SUVR rate of change derived as the slope from linear mixed-effect models in the same participants group as in **a**. The group of Aβ-negative controls spans both middle-aged and older individuals. In Supplementary Fig 1, we show individuals above 50 years old, as well as dividing the non-demented group into CU and MCI. Source data are provided as a Source Data file. Aβ beta-amyloid, AD Alzheimer's disease, CU cognitively unimpaired, MCI mild cognitive impairment, PET positron emission tomography, SUVR standardized uptake value ratio.

shown to be sensitive to early tau aggregates[32,33]. Insoluble Aβ aggregates were measured with Aβ-PET using [$^{18}$F]flutemetamol as the radiotracer. All PET data were parcellated in 200 cortical regions-of-interest (ROI) using the Schaefer brain atlas[34]. For analyses done across the 200 brain regions, only results surviving multiple comparisons from false discovery rate are reported. Concentrations of soluble p-tau were measured using p-tau217 in the CSF[35]. Accumulation of tau aggregates was quantified as the rate of change in tau-PET retention over time (SUVR/year) in each brain region separately, derived from linear mixed effect models. Cognitive decline was quantified as rate of change on cognitive scores (a cognitive composite score analogous to the PACC5 and MMSE) per year derived from linear mixed effect models. The main results focusing on early AD were also validated in the Alzheimer's Disease Neuroimaging Initiative (ADNI) database using [$^{18}$F]flortaucipir PET, CSF p-tau181 as well as [$^{18}$F]florbetapir or [$^{18}$F]florbetaben Aβ-PET ($n = 119$: 52 CU Aβ-negative, 36 Aβ-positive CU and 31 Aβ-positive MCI participants; all figures and extended results in Supplementary information). Given the smaller sample size in ADNI, only analyses where results could be detected with sufficient power calculated based on the BioFINDER-2 analyses were conducted.

In the BioFINDER-2 cohort, the baseline distribution of insoluble tau aggregates assessed via tau-PET recapitulated the AD-typical deposition in the medial and lateral temporal lobes in the controls and non-demented participants, and into lateral and medial parietal and lateral occipital regions at symptomatic AD stages (Fig. 1a and Supplementary Fig. 1 for CU and MCI separately). The longitudinal accumulation rate of tau aggregates (i.e., the annual tau-PET rate of change measured over a mean time of 2 years) was highest in temporal lobe regions in non-demented participants, and showed more extensive involvement mainly of medial parietal and lateral frontal regions in AD dementia (Fig. 1b and Supplementary Fig. 1). Quantitatively, focusing

on regions corresponding to a temporal meta-ROI encompassing medial and lateral parts of the temporal lobe - key regions of tau aggregates early in the disease process and approximating Braak stages I to IV -, the average annual SUVR rate of change was 0.7% in Aβ-negative controls, 3.3% in Aβ-positive non-demented individuals, and 9.0% in Aβ-positive patients with AD dementia. Between-group comparisons showing regions where accumulation of tau aggregates differ are also reported in Supplementary Fig. 2 for complementary description of regional differences of tau-PET rate of change.

In ADNI (Supplementary Table 1), the spatial pattern of the main regions with highest tau-PET binding at baseline and the highest rate of change in tau aggregates per year mirrored the patterns found in BioFINDER-2 (Supplementary Fig. 3). However, the magnitude of accumulation of tau aggregates was almost twice as small compared to BioFINDER-2.

### Increased soluble p-tau is the main modifier of tau aggregate accumulation rates in early AD

First, we tested whether either the level of local Aβ aggregates and/or the concentrations of soluble p-tau were most strongly associated with local increases in insoluble tau aggregates over time in early stages of AD using linear regression models in each of the 200 brain regions. Across Aβ-positive non-demented participants, regional Aβ was positively associated with greater accumulation of tau aggregates over time, most prominently in temporo-parietal regions (Fig. 2a). On average across the 82 regions surviving correction for multiple comparisons, the standardized estimate of Aβ was 0.26 (range 0.21 to 0.41, average $p$-values in those regions = 0.03). Levels of soluble p-tau was also positively correlated with accumulation of tau aggregates, but in contrast to regional Aβ, this was observed in nearly all brain regions ($n = 194$) and the effect sizes were substantially higher (Fig. 2a, average

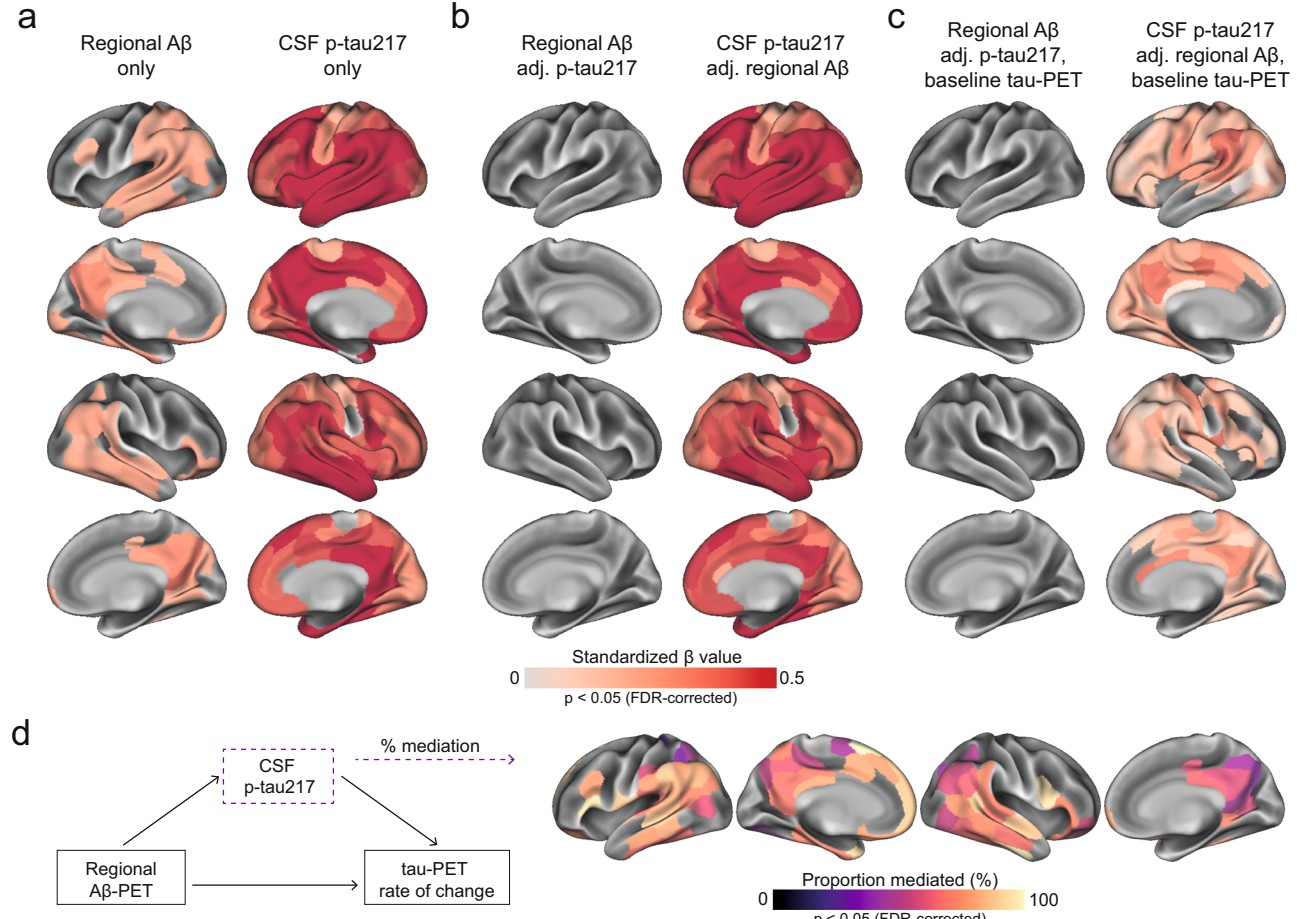

**Fig. 2 | Regional Aβ-PET and CSF p-tau217 associations with regional tau-PET [18F]RO948 rate of change in Aβ-positive non-demented participants.**
**a** Standardized beta coefficient of local Aβ-PET in regions where regional Aβ-PET flutemetamol SUVR (left column) relates to regional tau-PET rate of change, adjusting for age and sex. Right columns were derived from a similar model, but using CSF p-tau217 as predictor instead of Aβ-PET. **b** Standardized beta coefficient where local Aβ-PET (left column) and CSF p-tau217 (right column) is associated to regional tau-PET rate of change when including both biomarkers in the same model, adjusting for age and sex (tau PET rate of change ∼ regional Aβ-PET + CSF p-tau217 + age + sex). **c** Same depiction as in **b**, when additionally controlling for

regional baseline tau-PET SUVR. **d** Mediating effect of CSF p-tau217 on local Aβ-related accumulation of local tau aggregates. The mediation models were performed region-wise, and the percentage of the mediating effect are projected on the brains. Statistical details of the mediation models are shown in Supplementary Fig. 8. Significance of the mediation effect was tested using 1000 bootstrapping iterations. All regions shown on the brain are significant at $p < 0.05$ after FDR-correction from two-sided statistical tests. Source data are provided as a Source Data file. Aβ beta-amyloid, CSF cerebrospinal fluid, FDR false discovery rate, PET positron emission tomography, p-tau phosphorylated tau, SUVR standardized uptake value ratio.

standardized estimate: 0.47, range 0.19 to 0.71, average *p*-values <0.001). When using regional Aβ and soluble p-tau simultaneously as predictors (Fig. 2b), the widespread effect of CSF p-tau on the accumulation rate of tau remained virtually the same. In contrast, regional Aβ had almost no independent effect on tau accumulation after accounting for soluble p-tau. Of note, a similar effect of Aβ being no long significant after accounting for soluble p-tau was found on baseline tau aggregates (Supplementary Fig. 4). Furthermore, all key regions where soluble p-tau was most strongly associated with accumulation of tau aggregates over time remained significant even when additionally accounting for baseline levels of tau aggregates in each region (Fig. 2c), albeit with slightly lower estimates (average standardized estimate: 0.21, range: 0.05 to 0.44, average *p*-values = 0.007). These results were also consistent across different measures of Aβ, i.e. if using global Aβ load assessed by PET or the CSF Aβ42/40 ratio instead of regional Aβ aggregates in regression models (Supplementary Fig. 5). Further, investigating the same relations in the CU and MCI groups separately, the strong effect of soluble p-tau on subsequent tau aggregation rate, above the effect of Aβ, was clearly found in both groups (Supplementary Fig. 6). When further adjusting for local baseline tau aggregates, soluble p-tau also remained a significant

predictor in temporo-parietal regions in each group (Supplementary Fig. 6).

Lastly, based on the observed interplay between local Aβ and soluble p-tau on the increased rate of tau aggregates accumulation, we formally tested the mediating effect of soluble p-tau on Aβ-related tau-PET rate of change at the regional level. Figure 2d summarizes the proportion of mediating effect of soluble p-tau on all regions surviving multiple comparisons ($n = 82$), which averaged to 70% across all regions. Overall, the results indicate that the effects of local Aβ aggregates on the accumulation rate of insoluble tau aggregates over time is largely mediated by increased concentrations of soluble p-tau.

In ADNI, as in BioFINDER-2, soluble p-tau, measured with CSF p-tau181, was the main factor related to regional accumulation of tau aggregates over time (Supplementary Fig. 7). Analyses were conducted using two Aβ measures: the global centiloid (CL) score since two different PET Aβ tracers are used in ADNI, and CSF Aβ42. In both cases, soluble p-tau remained significant when accounting for Aβ and baseline tau-PET SUVR in temporo-parietal regions, although the regional pattern was more restricted than in BioFINDER-2 (Supplementary Fig. 7).

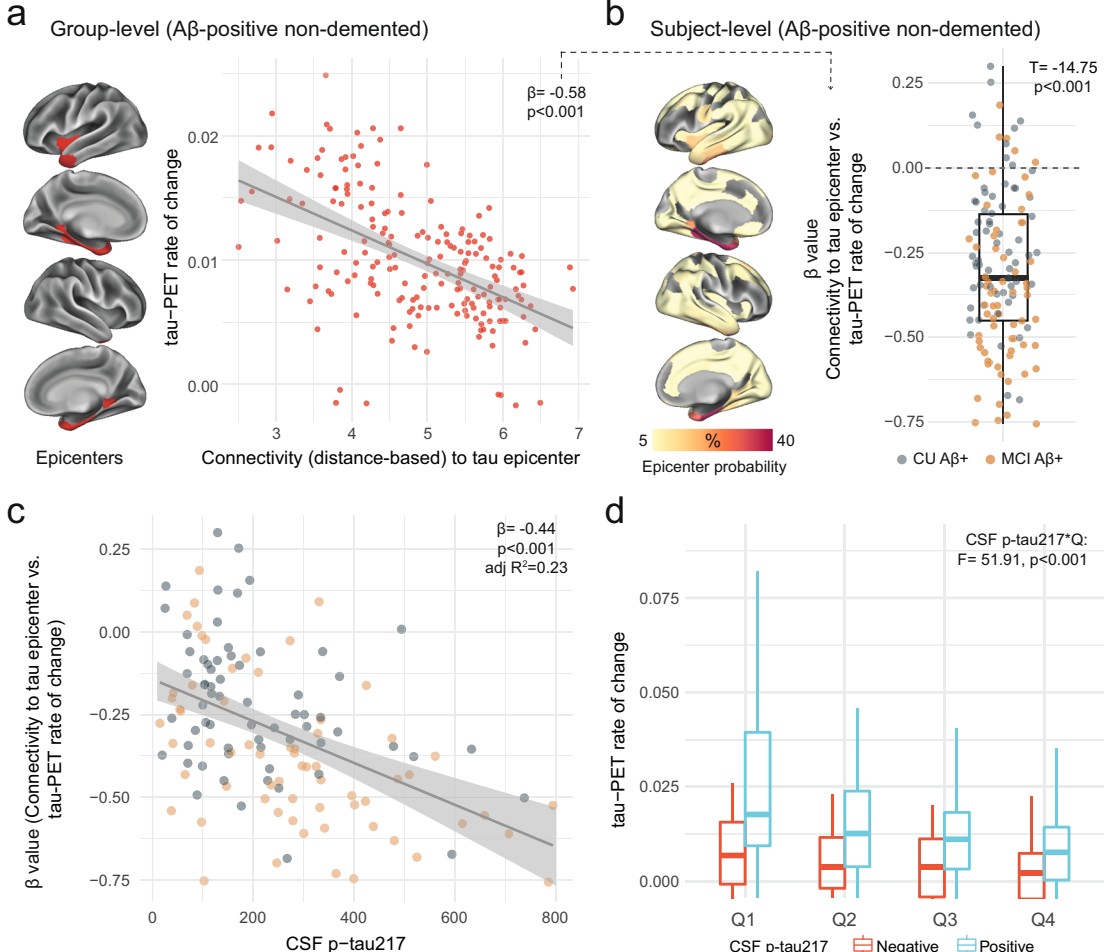

**Fig. 3 | Individualized connectivity-based associations of tau-PET rate of change over time and CSF p-tau217 in Aβ-positive non-demented participants.** **a** Group-level analysis showing how connectivity to the tau epicenters (projected on the glass brains) relates to tau-PET rate of change across the whole brain. Each dot represents a brain region. Regions more strongly functionally connected to the epicenters have greater rate of tau-PET accumulation. **b** Repeating the same approach depicted in **a** at the individual level, the values on the glass brains represent the percentage that each region is classified as an epicenter. The box plot shows the individual β-value ($n = 130$) from the correlation between tau-PET rate of change and connectivity-based distance to epicenters across all brain regions. **c** Scatter plot of the associations between CSF p-tau217 and the β-values of epicenter connectivity to tau-PET rate of change. Each dot represents an individual ($n = 130$). The expected negative association suggests that higher CSF p-tau217 is associated with the overall pattern of tau-PET change in more functionally connected regions to epicenters. **d** Average tau-PET rate of change across all participants ($n = 130$) in regions split into quartiles based each region's connectivity to the tau epicenters (Q1 represents top 25% regions with strongest functional connectivity to the epicenters, etc.). All linear regressions performed were two-sided, without adjustment for multiple comparisons and error bands correspond to the 95% confidence interval. In all box plots, the box limits represent the interquartile range and the line depicts the median value. Aβ beta-amyloid, CSF cerebrospinal fluid, PET positron emission tomography, p-tau phosphorylated tau, Q quartile.

## Soluble p-tau levels relate to connectivity-based accumulation of tau aggregates in early AD

Another important modifier of regional tau accumulation is the functional architecture of the brain. We therefore investigated whether the association between connectivity-based accumulation of tau aggregates was influenced by the concentration of soluble p-tau. Briefly, after defining participant-specific tau-PET epicenters (i.e., the top 10 regions with the highest tau-PET SUVR probability from Gaussian-mixture modeling at baseline), accumulation of tau aggregates in the remaining 190 regions was correlated to functional connectivity strength to the epicenters[27,36]. We first applied the model at the group-level, where we found greater accumulation rates of insoluble tau aggregates in regions more strongly connected to the epicenters (shorter distance-based connectivity) in early AD (Fig. 3a). Next, we applied this model at the individual level, defining tau epicenters for each participant. We found that higher accumulation rates of tau aggregates were observed in regions that showed strongest connectivity to the tau epicenters defined on baseline tau-PET, as

evidenced by negative β-values (i.e. reflecting the association between connectivity to the epicenters and tau aggregates accumulation across all non-epicenter ROIs; Fig. 3b).

Next, given the association between soluble p-tau and the accumulation of tau aggregates shown in Fig. 2, we hypothesized that higher soluble p-tau concentrations would relate to a stronger association between connectivity to the epicenters and the accumulation rates of tau aggregates across brain regions. In all non-demented Aβ-positive participants, we found that participants with higher soluble p-tau levels had a stronger association between connectivity to epicenter and tau-PET rate of change in non-epicenter ROIs (β-value), while accounting for global Aβ, age and sex (Fig. 3c). This association also survives adjustment for baseline tau-PET (β = −0.23, p = 0.04). The same associations were found in CU and MCI participants alone (Supplementary Fig. 9), suggesting that this effect is not only due to the MCI participants who tend to show greater association between connectivity to epicenter and tau-PET rate of change (more negative β-values on Fig. 3b). Analyzing the data in a complementary way, we

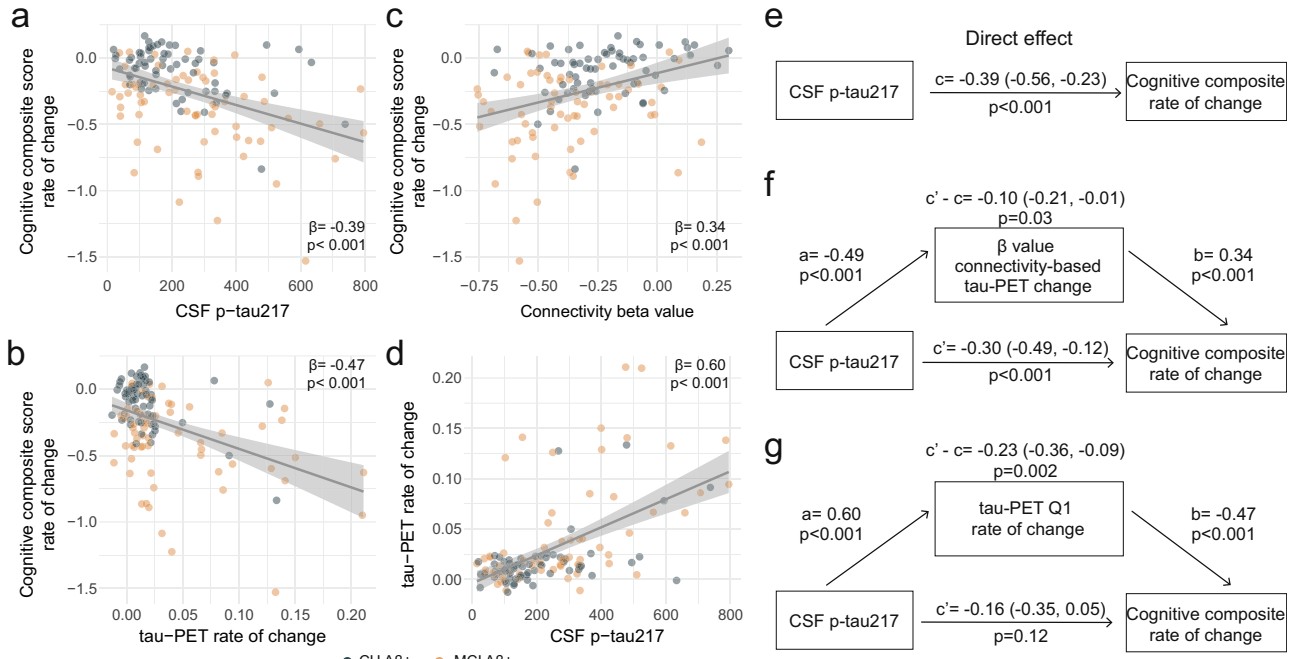

**Fig. 4 | Tau aggregates accumulation mediates associations between CSF p-tau217 and cognitive decline in Aβ-positive non-demented participants.**
**a–d** Scatter plots of associations relevant to subsequent mediation analyses, beta coefficients from linear regressions adjusting for age, sex and education are reported. **a** Association between CSF p-tau217 and rate of change on the cognitive composite score. **b** Association between tau-PET rate of change in Q1 and rate of change on the cognitive composite score. **c** Association between β-value from the correlation between tau-PET rate of change and connectivity-based distance to epicenters across all brain regions and rate of change on the cognitive composite score. **d** Association between CSF p-tau217 and tau-PET rate of change in Q1. **e–g** Mediation analysis of the relationship between CSF p-tau217, measures of tau-PET and cognitive decline measured as the rate of change on the cognitive composite score. The direct effect (**c**) of CSF p-tau217 on cognitive decline is shown in **e**.

Analyses are shown with β-value based on connectivity and tau-PET change (**f**), and tau-PET rate of change in Q1 (**g**) as mediators. The mediated effect is designated *c-c'*. The remaining effect of CSF p-tau217 on cognitive decline after adjusting for the mediator is designated *c'*. 95% confidence intervals derived from 1000 simulations are reported in parentheses. The direct effect of CSF p-tau217 on the mediator is *a*, and the direct effect of the mediator on cognitive decline is *b*. The β-value based on connectivity and tau-PET change (**f**) and tau-PET rate of change (**g**) mediated the relationship between CSF p-tau217 and cognitive decline. To facilitate model comparisons, all models use continuous standardized (*z*-score) data for variables of interest. All linear regressions performed were two-sided, without adjustment for multiple comparisons and error bands correspond to the 95% confidence interval. Aβ beta-amyloid, CSF cerebrospinal fluid, PET positron emission tomography, p-tau phosphorylated tau, Q quartile.

measured the rate of tau aggregates accumulation in regions split into quartiles defined at the individual level, i.e. average tau-PET rate of change in top 25% regions with the greatest connectivity to tau epicenters as quartile 1 (Q1), up to quartile 4 (Q4, regions with the lowest connectivity to tau epicenters). Repeated measures ANOVA revealed an interaction of soluble p-tau levels and quartiles ($F = 51.9$, $p < 0.001$), suggesting the importance of soluble p-tau and connectivity-based regions on the accumulation rates of insoluble tau aggregates (Fig. 3d). This effect was particularly evident Q1, which showed the greatest effect size between CSF p-tau-positive and p-tau-negative participants (Cohen's d = 0.70 vs. 0.57 to 0.61 in Q2 to Q4).

Results were also validated in the Aβ-positive non-demented participants from ADNI (Supplementary Fig. 10). The overall connectivity-based association with tau-PET rate of change (β-value) was related to the levels of soluble p-tau (standardized coefficient = −0.24, $p = 0.01$), there was an interaction between soluble p-tau and connectivity-based tau aggregates accumulation by quartiles ($p = 0.01$).

**Accumulation of insoluble tau aggregates mediates the associations between soluble p-tau and cognitive decline in early AD**
With soluble p-tau being related to local accumulation of tau aggregates and connectivity-mediated tau accumulation, we then investigated how those different tau measures related to cognitive decline in early stages of AD (i.e. Aβ-positive non-demented participants). We focused on tau-PET rate of change in regions most strongly connected to the epicenters (i.e. Q1), to capture individualized level of tau

aggregates accumulation, and all results were also validated using a commonly used temporal meta-ROI (see Supplementary Fig. 11). All different tau-related measures (CSF p-tau217, tau-PET rate of change in Q1 and the β-value of connectivity-based tau-PET rate of change) were associated with cognitive decline as measured by annual change on a cognitive composite score over time, designed to capture cognitive changes in the earliest AD stages (Fig. 4a–c). Given the observation that soluble p-tau increases early in AD and prior to the formation of neocortical insoluble tau aggregates, we then tested whether measures related to the accumulation rate of insoluble tau aggregates mediated the association between soluble p-tau concentrations and subsequent cognitive decline. First, the strength (β-value) of the association between accumulation rate of tau aggregates and functional connectivity to tau epicenters mediated 25% of the association between soluble p-tau and cognitive decline (Fig. 4f), suggesting a partial mediating effect of faster connectivity-based tau aggregates accumulation. Second, the accumulation rate of tau aggregates in the regions most connected to the subject-specific tau epicenter (Q1) mediated 60% of the association between soluble p-tau concentrations and the rate of cognitive decline (Fig. 4g). In sensitivity analyses, using the slopes of MMSE scores to measure cognitive decline, we found very similar mediating effects of tau aggregation (for mediating effects of the connectivity-based β-value: *c'-c* = −0.09 [95% CI −0.19, 0.00], $p = 0.04$; for mediating effect of the tau-PET rate of change in Q1: *c'-c* = −0.25 [−0.39, −0.13], $p < 0.001$). The same analyses were also repeated splitting the non-demented group into Aβ-positive CU and MCI separately. We observe a full mediation effect of tau-PET rate of

change, either in Q1 or in the temporal meta-ROI between soluble p-tau and cognitive decline on the cognitive composite score in CU (c'- c = −0.21 [−0.40, −0.06], p = 0.002), and on decline on MMSE in MCI (c'-c = −0.24 [−0.46, −0.04], p = 0.01), see Supplementary Fig. 12 for detailed statistics. The only analysis that we could not replicate in individual groups was the mediating effect of the β-value of connectivity-based tau-PET change (Fig. 4f), which was also of smaller magnitude in the full group of non-demented participants. Taken together, the results indicate that higher concentrations of soluble p-tau are associated with cognitive decline in early AD, which is mediated by increased accumulation rates of insoluble tau aggregates.

Based on effect sizes determined in BioFINDER-2, sample sizes between 75 and 82 participants would have been needed to assess effects of soluble p-tau on cognitive decline in ADNI. Since only 28 participants had longitudinal cognitive assessments, this set of analyses was not conducted in ADNI.

### Distinct associations between soluble p-tau and accumulation of tau aggregates in AD dementia

All previous analyses focused on non-demented participants, to study the effects of soluble p-tau and connectivity on tau-PET rate of change in early stages of AD. To investigate the full AD clinical continuum, we repeated the main analyses focusing on Aβ-positive individuals with AD dementia. Our motivation to focus on AD dementia patients separately was based on the plateau-phase reached by soluble p-tau species in more advanced disease stages (Fig. 5a with CSF Aβ42/40 to include the full sample since AD dementia patients do not undergo Aβ PET, and Supplementary Fig. 13 with Aβ PET in a restricted sample). In Aβ-positive non-demented participants, higher soluble p-tau concentrations were associated with faster tau aggregates accumulation, above levels of Aβ and baseline tau aggregates. In contrast, in AD dementia, soluble p-tau concentrations were not associated with the accumulation rates of tau aggregates in any region after adjusting for baseline levels of local tau aggregates (Fig. 5b). The expected negative association between greater regional accumulation rates of tau aggregates and connectivity to the tau epicenters across the brain was present at the dementia stage, but the strength of this association (β-value) was not related to soluble p-tau levels (Fig. 5c). Similarly, soluble p-tau concentrations did not have a direct effect on cognitive decline (Fig. 5d). Rather, the accumulation rate of tau aggregates over time was most associated with cognitive decline at this stage of the disease (Fig. 5e in Q1 and β = −0.36, p < 0.001 for association in the temporal meta-ROI). Further, soluble Aβ levels, measured as the CSF ratio of Aβ42/40, were not related to the accumulation of tau aggregates (β = 0.07, p = 0.55 for tau-PET rate of change in Q1) or cognitive decline (β = −0.21, p = 0.10 for MMSE slope). Overall, these results suggest that the accumulation rate of tau aggregates and cognitive decline seem independent of soluble p-tau concentrations in the dementia stage of the disease.

## Discussion

The major aim of the present longitudinal biomarker study was to assess the relations between soluble p-tau concentrations and Aβ-related accumulation of insoluble tau aggregates over time in AD. In a large sample of Aβ-positive non-demented individuals of the BioFINDER-2 cohort, we found first, that soluble CSF p-tau fully mediated the effects of regional Aβ on the subsequent accumulation rate of tau aggregates measured with longitudinal tau-PET, and second, that elevated soluble p-tau was associated with faster accumulation of tau aggregates across functionally connected brain regions. Importantly, these findings were replicated in the ADNI cohort. Third, we found that elevated soluble p-tau concentrations were associated with faster cognitive decline in early stages of AD, which was mediated by faster accumulation rates of tau aggregates. All findings in non-demented group also even held when analyzing CU and MCI groups

separately (Supplementary Figs. 6, 8 and 12). However, soluble p-tau concentrations, that plateau at the late-stage of Aβ accumulation, were no longer associated with local accumulation of tau aggregates or cognitive decline in patients with AD dementia. At this advanced stage, baseline levels of tau aggregates predicted subsequent accumulation of tau aggregates in the same brain regions[37], and we further showed that tau aggregation rates related to greater cognitive decline. These results suggest a close link between the accumulation of tau aggregates and clinical deterioration once Aβ and p-tau levels have plateaued in late-stage AD. Taken together, these findings are congruent with the hypothesis that Aβ-induced soluble p-tau concentrations might play a key role in initiating the aggregation and connectivity-mediated accumulation of tau pathology in early-stage AD and that local tau seeding and self-replication predominate once soluble p-tau concentrations have plateaued in AD dementia. This might have implications for clinical trials, since drugs reducing soluble p-tau concentrations (like anti-Aβ therapies or genetic anti-tau treatments) may be promising therapeutic strategies to prevent further accumulation and spread of tau aggregates and cognitive decline in early stages of AD. On the other hand, during the dementia stage of AD, directly targeting the local tau aggregates might prove to be more adequate than targeting soluble p-tau species or Aβ.

To gain a better understanding of AD pathogenesis, it is important to elucidate how key biomarkers relate to longitudinal changes along the disease continuum. Here we found that soluble p-tau was a key biomarker in the Aβ-related accumulation of tau aggregates and of cognitive decline over time in early AD. Importantly, the effect of CSF p-tau on increases in tau-aggregates survived adjustment for baseline levels of Aβ and tau aggregates in the same brain region, suggesting a unique contribution of soluble p-tau in subsequent tau aggregation, and not only on baseline levels of tau aggregates[38,39]. Moreover, soluble p-tau mediated the association between Aβ and accumulation rate of subsequent tau aggregates, expanding on previous cross-sectional results[7,10]. As such, Aβ-related increases in soluble p-tau may be a key initial step in the Aβ cascade that determines the accumulation of aggregated tau pathology, and thereby leading to faster cognitive decline in early AD, and this cascade of events was even found in the preclinical stage of the disease. This finding expands on previous studies linking longitudinal tau-PET and cognitive decline[40,41], by highlighting the importance of soluble p-tau in early stages of the disease.

From a pathophysiological point of view, Aβ has been shown to trigger increased synthesis, hyperphosphorylation and secretion of tau proteins from neurons, which leads to elevated interstitial p-tau concentrations, that pass into CSF and blood plasma where it can be detected in vivo using biomarkers[1,9]. Preclinical studies provide strong support for neuronal activity as a putative link between Aβ and tau secretion and accumulation. Aβ has consistently been shown to induce a hyper-excitatory shift in neuronal activity[21,42,43], and this neuronal hyperexcitability may then induce increased p-tau secretion and subsequent spreading, with neuronal tau secretion being drastically enhanced by elevated neuronal activity[20,23,24]. This view is supported by our result showing that elevated soluble p-tau was associated with faster accumulation of tau aggregates across functionally connected brain regions. While accumulation of tau aggregates in a given brain region was related to its functional connectivity to tau epicenters[27,36,44], the accumulation of tau aggregates from local epicenters to connected regions was enhanced in the face of abnormal soluble p-tau. Overall, both connectivity and soluble p-tau were important factors of the rate of tau aggregates accumulation. These findings suggest that Aβ-related soluble p-tau increases could be key prerequisite for the expansion of tau aggregates across functionally connected brain regions in early AD.

Importantly, the results described above only apply to early pre-dementia AD stages and not to the dementia stage of AD. We found that the key role of soluble p-tau in the accumulation process of tau

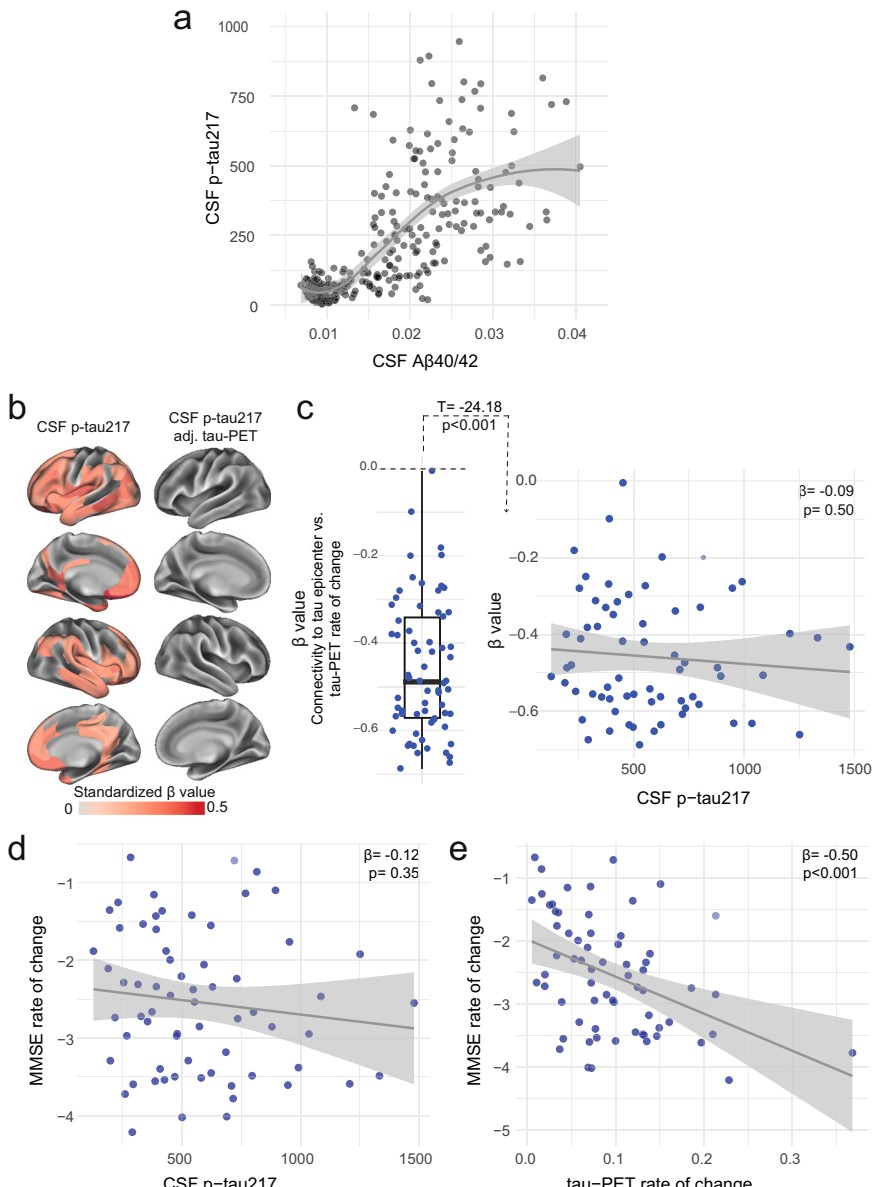

**Fig. 5 | Distinct associations between CSF p-tau217 and tau-PET rate of change in the AD dementia stage. a** Associations between the ratio of CSF Aβ40/42 from Elecsys and CSF p-tau217 across the AD continuum. CSF Aβ40/42 is shown here instead of PET to include the entire BioFINDER-2 cohort, since AD dementia patients do not undergo Aβ-PET. A similar association with Aβ-PET SUVR is shown in a restricted sample without AD dementia patients in Supplementary Fig. 13. The association is nonlinear and flatten at high Aβ load. **b** CSF p-tau217 alone was mildly associated with tau-PET rate of change (left column), which was not the case when additionally adjusting for regional baseline tau-PET SUVR (right column). **c** Box plot showing the expected negative β-value from the correlation between tau-PET rate of change and connectivity-based distance to epicenters across all AD patients, but

this β-value was not related to CSF p-tau217 levels. Scatter plots of associations between CSF p-tau217 (**d**) and tau-PET rate of change in Q1 (**e**) and cognitive decline, as measured by MMSE rate of change. Beta coefficients from linear regressions adjusting for age, sex and education are reported. The rate of tau aggregates accumulation was related to cognitive decline, which was not the case for CSF p-tau217. All linear regressions performed were two-sided, without adjustment for multiple comparisons and error bands correspond to the 95% confidence interval. Aβ beta-amyloid, AD Alzheimer's disease, CSF cerebrospinal fluid, MMSE Mini-mental state examination, PET positron emission tomography, p-tau phosphorylated tau.

aggregates was no longer observed in patients with AD dementia. This can likely be explained by a plateau stage of Aβ aggregates and soluble p-tau levels occurring in more advanced disease stages[10,45]. In our data, soluble p-tau levels had a minor effect on further accumulation of tau aggregates and connectivity-based tau spreading in AD dementia, and only the tau aggregatse accumulation rate was associated with cognitive decline (and not p-tau levels).

The main strength of this study is the integration of cross-sectional and longitudinal fluid and neuroimaging biomarkers of AD pathology and longitudinal cognitive measures across the clinical

spectrum of AD. There are also several limitations. First, although we are expanding on previous studies that were limited by shorter follow-up or smaller sample sizes[46–48], a longer follow-up time of tau-PET would increase the rate of tau aggregates accumulation in earlier stages of AD, which span many years. The correlational study design precludes us from making conclusive statements on soluble p-tau as the main driver of subsequent tau aggregation, but investigating factors related to tau-PET rate of change brings novel insight into AD pathophysiology in vivo. Second, there were some differences between the biomarkers and cognitive data between BioFINDER-2 and

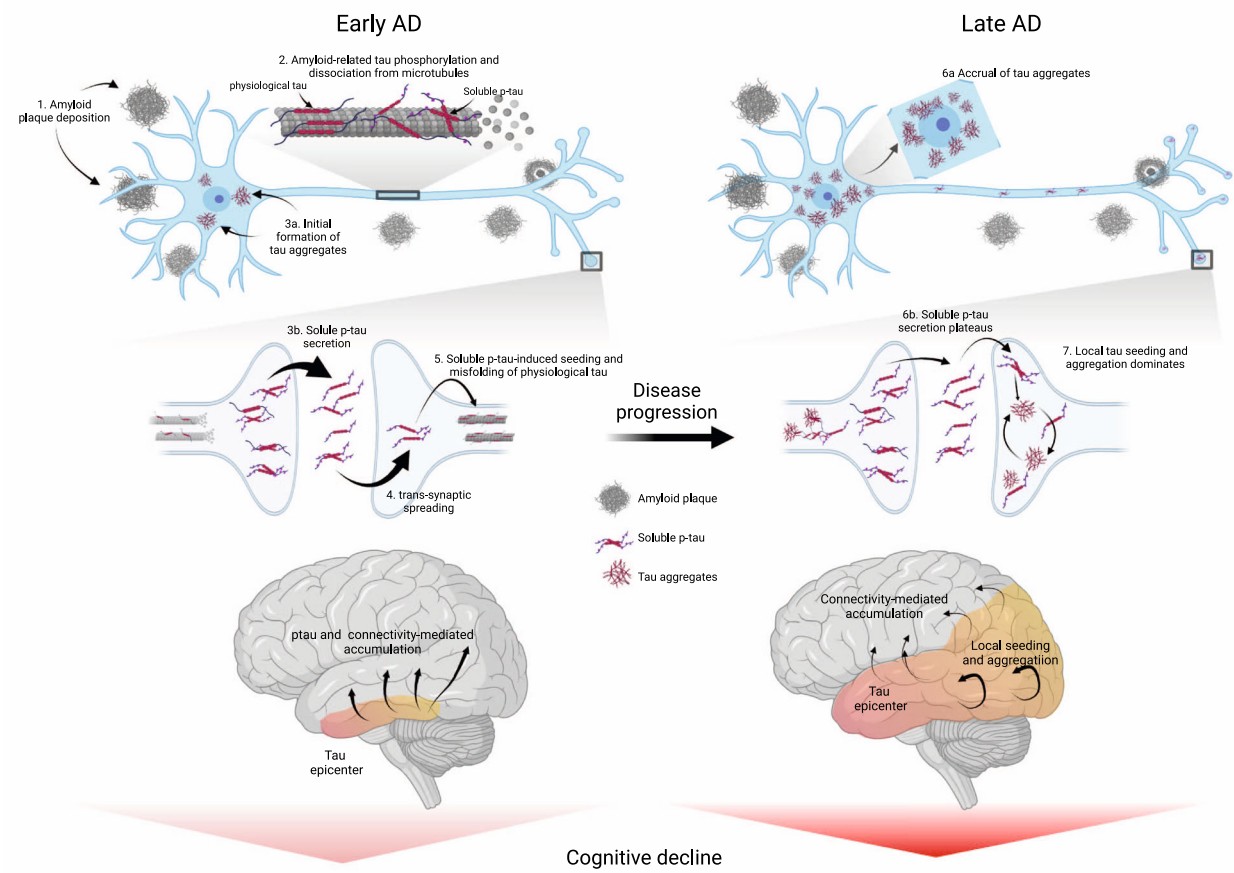

**Fig. 6 | Proposed model of tau pathology accumulation in Alzheimer's disease.** In early AD, soluble or insoluble Aβ triggers increased concentrations and secretion of soluble p-tau, followed by post-synaptic uptake of p-tau seeds that lead to tau misfolding and aggregation. In late stages of the disease, when soluble p-tau concentrations have reached a plateau, local tau aggregates rather dominate in driving further local tau aggregation. Figure created with BioRender.com. Aβ beta-amyloid, AD Alzheimer's disease, p-tau phosphorylated tau.

ADNI. In BioFINDER-2 the focus was on CSF p-tau217, having shown greatest association with Aβ[12,49], but the main results could still be validated in ADNI where only CSF p-tau181 was available. Unfortunately, cognitive decline could not be investigated in ADNI due to the small number of participants with longitudinal cognitive data, but we showed the robustness of these results in BioFINDER-2 by using two cognitive scores and different regions of interest of tau aggregates accumulation. Lastly, we acknowledge that functional connectivity is an indirect measure of brain activity, but still has been shown to be related to mono- and polysynaptic pathways[50].

In conclusion, reconciling both early and later stages of the disease, we propose an integrative model of how Aβ fibrils, soluble p-tau concentrations, functional connectivity, accumulation rates of insoluble tau aggregates and cognitive decline are interrelated across the entire clinical continuum of AD (Fig. 6). In this model, initially the Aβ-related increase of p-tau seeds are taken up by neurons to initiate the misfolding and aggregation of tau in a given brain region, with more substrate (i.e. soluble p-tau) for aggregation leading to more rapid local tau aggregation. Later on, local self-replication and accumulation of misfolded tau aggregates may take over once a critical threshold of tau aggregates is reached. This distinct-stage process also aligns with recent evidence of trans-neuronal tau spreading being critical for the initial expansion of tau pathology, but local replication of tau pathology driving accumulation of tau aggregates in later stages[51]. Stemming from this model, our work has potential implications for therapeutic approaches. We propose that targeting Aβ fibrils and soluble p-tau in early AD may be a promising strategy to slow the formation of tau aggregates, and thereby to preserve cognitive abilities for a longer period. Preliminary evidence from recent phase 2 and 3 clinical trials suggest that anti-Aβ therapies promptly reduce soluble p-tau concentrations in both CSF and plasma, reenforcing the close relationship between Aβ pathology and increases in extracellular p-tau levels[52–54]. Applying the same principles to anti-tau therapies (see ref. 55 for a review), we speculate that approaches reducing tau production and phosphorylation (e.g. antisense oligonucleotides, post-translational modification modulators, passive immunotherapies) would be most effective early on. On the other hand, in late stages of the disease, when local tau aggregation is the predominant pathway leading to tau accumulation, targeting tau aggregates may exert stronger clinical benefits. As such, it is possible that therapeutics acting on tau aggregation inhibition and active clearance of aggregates might be more beneficial in the dementia stage. Our findings may be a starting point for future precision-medicine interventions that target the dominant pathway that determines tau aggregation, which are Aβ and soluble p-tau increases in early AD and tau aggregates in late AD.

## Methods

### Participants - BioFINDER-2 cohort

The main cohort of interest included participants from the ongoing prospective Swedish BioFINDER-2 cohort (NCT03174938, http://www. biofinder.se/), which has been approved by the ethics committee at Lund University. All participants were recruited at Skåne University Hospital and the Hospital of Ängelholm, Sweden and the cohort covers the full spectrum of AD, ranging from adults with intact cognition or

subjective cognitive decline, mild cognitive impairment (MCI), to dementia. Informed consent was obtained from all participants, and they were compensated for each study visit. The main inclusion criteria, as described previously[15], were to be 40 years and older, being fluent in Swedish, having Mini-Mental State Examination (MMSE) between 27 and 30 for cognitively unimpaired (CU) participants, between 24 and 30 for MCI, and equal to or above 12 for AD dementia patients. Exclusion criteria were having significant unstable systemic illness, neurological or psychiatric illness, significant alcohol or substance misuse, or refusing lumbar puncture or neuroimaging. MCI diagnosis was established if participants performed below 1.5 standard deviation from norms on at least one cognitive domain from an extensive neuropsychological battery examining verbal fluency, episodic memory, visuospatial ability, and attention/executive domains. AD dementia diagnosis was determined on the criteria for dementia due to AD from the Diagnostic and Statistical Manual of Mental Disorders Fifth Edition and if positive on Aβ biomarkers based on the updated NIA-AA criteria for AD[56]. The study was approved by the Regional Ethics Committee in Lund, Sweden. All participants gave written informed consent to participate. All data for the current study was acquired between April 2017 and July 2021, and participants included needed to have CSF p-tau217 measurements and longitudinal tau-PET and cognitive data. After quality control of tau-PET (described below), the final sample included the following participants: 204 CU Aβ-negative, 65 CU Aβ-positive, 65 MCI Aβ-positive patients and 66 AD dementia patients. Follow-up time in tau-PET ranged from 0.73 to 2.5 years. CU participants are followed up every two years, while MCI and AD patients are seen annually.

### Participants - ADNI
ADNI is a multi-site study launched in 2003 as a public-private partnership. The primary goal of ADNI has been to test whether serial MRI, PET, other biological markers, and clinical and neuropsychological assessment can be combined to measure the progression of MCI and early Alzheimer's disease. For up-to-date information, see www.adni-info.org. ADNI3 (NCT02854033) data was used in the current study and the study was approved by all relevant ethic boards related to ADNI (See Reporting Summary). Informed consent was obtained from all participants, and they were compensated for their participation. We selected participants who had longitudinal tau-PET, Aβ-PET (i.e. florbetaben or florbetapir) and CSF p-tau181 available in the ADNI database as of July 2021, which amounted to 122 participants out of the 777 individual participants with tau-PET. Out of those, only 3 were diagnosed with AD dementia and given the too small sample size, were not included in the study. The final sample of 119 thus included CU and MCI participants. Clinical status was assessed by ADNI: CU participants had a MMSE above 24, a Clinical Dementia Rating (CDR) score of 0, and were not depressed; participants with MCI had a MMSE above 24, a CDR of 0.5, objective memory-impairment based on education-adjusted Wechsler Memory Scale II and preserved activities of daily living.

### Image acquisition and processing - BioFINDER-2 cohort
MRI was performed using a Siemens 3T MAGNETOM Prisma scanner (Siemens Medical Solutions). Structural T1-weighted MRI images were acquired from a magnetization-prepared rapid gradient echo (MPRAGE) sequence with 1 mm isotropic voxels. PET images were acquired on digital GE Discovery MI scanners. For tau-PET, acquisition was done 70–90 min post injection of ~370 MBq [18F]RO948. For Aβ-PET, acquisition was done 90–110 min post injection of ~185 MBq [18F]Flutemetamol. Images were processed according to our pipeline described previously[57]. Briefly, PET images were attenuation corrected, motion corrected, summed and registered to the closest T1-weighted MRI processed through the longitudinal pipeline of FreeSurfer version 6.0. Standardized uptake value ratio (SUVR) images were created using

the inferior cerebellar gray matter as the reference region for [18F]RO948[58], and the pons for [18F]Flutemetamol. PET SUVR images were then warped non-linearly to the Montreal Neurological Institute (MNI152) template space using Advanced Normalization Tools (ANTs) version 2.3.1[59] registration parameters of the T1-weighted MRI scan to the MNI152 template.

The cutoff for Aβ-positivity was 0.53 SUVR, defined from Gaussian mixture modeling (GMM) on a neocortical global Aβ region of interest (prefrontal, lateral temporal, parietal, anterior cingulate, and posterior cingulate/precuneus), as used previously[15]. This cutoff was used to classify CU and MCI participants, and a cutoff from CSF was used in AD dementia patients.

[18F]RO948 signal can be affected by off-target binding in the skull and meninges[33]. Given that the analyses focused on whole-brain cortical regions, we excluded a few participants with high levels of skull/meningeal binding, using a similar approach as described previously[60]. We calculated the SUVR ratio of a meningeal/skull mask to the average SUVR from GM, WM and CSF masks. Participants with a ratio above 1.75 were excluded, the latter having more than 1.75 times greater binding in the off-target region than in the brain. After such quality control, 12 CU Aβ-negative, 4 CU Aβ-positive, 2 MCI Aβ-positive participants and 1 AD dementia patient with longitudinal tau-PET were excluded from the final sample.

### Image acquisition and processing - ADNI
Details on acquisition procedures of PET and MRI images in the ADNI cohort can be found elsewhere (http://adni.loni.usc.edu/methods/documents/). Briefly, structural MRI data was acquired on 3T scanners, using 3D T1-weighted MPRAGE sequences with 1 mm isotropic voxel-size. PET data was acquired at standardized time-intervals post injection of 18F-labeled tracers. Flortaucipir was used for tau-PET and images were acquired 75–105 min post-injection. For Aβ-PET, florbetapir (image acquisition 50–70 min post-injection) and florbetaben (starting in ADNI3, image acquisition 90–110 min post-injection) were used. All image processing was done locally. Images were realigned, averaged, resliced to 1.5 mm$^3$, and smoothed to a resolution of 8 mm$^3$ FWHM. Each PET image was registered to the closest structural T1-weighted MRI image. SUVR images were created using the inferior cerebellum as the reference region for flortaucipir, and the whole cerebellum for florbetapir and florbetaben. Each T1-weighted image was also registered to the MNI template space using ANTs version 2.3.1. Applying these registration parameters, SUVR PET images were then also registered to the MNI space for further analyses.

Aβ positivity was taken from the thresholds established by the ADNI PET core. The cutoff for positivity from a global composite cortical region referenced to the whole cerebellum was 1.11 SUVR for florbetapir and 1.08 for florbetaben[61]. Further, to be able to analyze all participants together despite two Aβ tracers being used, global Aβ SUVR were converted to the Centiloid scale[62]. As such, analyses with Aβ-PET in ADNI were done based on the global Aβ centiloid score and not at the regional level.

### Cerebrospinal fluid markers
In BioFINDER-2 CSF p-tau217 was measured using immunoassays on the Meso-Scale Discovery platform developed by Eli Lilly as described previously[35,63]. Dichotomization into CSF p-tau217 positivity was determined at a cutoff of 114.4 pg/ml, determined from GMM on measures from the whole BioFINDER-2 baseline data ($n = 943$). Note that this threshold is virtually the same as when using 2 standard deviations from the mean of CU Aβ-negative participants (111.4 pg/ml). CSF Aβ42 and Aβ40 were measured using the Elecsys immunoassays (Roche Diagnostics)[13]. A pre-established cutoff of 0.08 on the CSF Aβ42/40 ratio was used to define Aβ-positivity[15]. AD patients did not undergo Aβ-PET, and thus for this group only, analyses were restricted to CSF Aβ42/40.

In ADNI CSF p-tau181 was measured at the University of Gothenburg using Elecsys immunoassays (Roche Diagnostics) and CSF p-tau217 was unavailable. CSF p-tau positivity was determined based using pre-established cut-offs at 21.8 pg/ml[13].

## Neuropsychological measures

In BioFINDER-2, cognitive decline was also investigated. We calculated a cognitive composite score analogous to the Preclinical Alzheimer's Cognitive Composite 5 (PACC5) as a measure to capture early cognitive decline[64,65]. The tests included were the MMSE, ADAS-cog delayed recall, Trail-Making Test version A and Category Fluency of animals. The original PACC includes two measures of memory recall (Logical Memory and the Free and Cued Selective Reminding Test). Given that in our cohort only one memory score was available, ADAS-cog delayed recall was assigned a double weight to maintain the same proportion of memory as in the original composite score as done previously[66]. Further, the Digit Symbol Substitution Test used in the original PACC was replaced by Trail-Making Test A, to avoid missing values, as not all MCI patients completed the Symbol Digit Modalities Test. All tests were z-scored based on the mean and standard deviation of CU Aβ-negative participants over 50 years old, and then averaged to generate the cognitive composite score. Participants had two to four cognitive assessments.

To calculate change in cognition over time, linear mixed effect models with random slope and intercept were fitted with the cognitive composite scores as the dependent variable and time in years from baseline as the independent variable. The slope of each participant from those models then represented rate of change on cognition per year. The same approach was applied to MMSE scores only, to have an alternate measure of cognitive change over time.

## Regional measures and rate of change of PET

In both cohorts, all PET (Aβ and tau) images were parcellated into 200 regions corresponding to the widely used Schaefer functional atlas, openly available on nilearn (nilearn.datasets.fetch_atlas_schaefer_2018)[67]. Average SUVR were extracted from these 200 parcels covering the entire neocortex, after masking each ROI with a gray matter mask to ensure that the final values to be used in analyses were least contaminated by binding from the white matter or CSF.

To calculate the rate of change in tau-PET over time, linear mixed effect models with random slope and intercept were fitted for each brain region, with tau-PET SUVR as the dependent variable and time in years from the baseline scan date as the independent variable. Participants had two to four tau-PET scans (see Table 1). The slope of each participant from those models then represented tau-PET SUVR change per year, referred to as rate of tau aggregates accumulation.

## Defining tau-PET epicenters

We defined tau-PET epicenters at the individual level from baseline tau-PET data, representing the regions with the highest probability of being abnormal using GMM. Given the skewed distribution of tau-PET, with many participants and many brain regions not exhibiting high levels of tau aggregates, we used GMM to separate target binding from non-specific binding. We fitted 2-distribution GMM on each brain region to separate both signals and extracted the probability to fall on the right-most distribution ("high" tau-PET distribution) for each participant[27,28,38]. Since this right-most distribution likely reflects abnormal tau-PET signal, the GMM probability represents the probabilistic measure of tau positivity, without the need to use a priori thresholds. The GMMs were fitted on the whole BioFINDER-2 baseline tau-PET data ($n = 934$), to have distributions corresponding as best as possible to the full range of tau-PET SUVR values across the AD spectrum. In ADNI, GMM were fitted on the sample with longitudinal tau-PET. A few parcels in the somatomotor regions with low tau-PET SUVR showed almost same fit between one or two distributions. They were

excluded from the epicenter selection given that we could not clearly separate high- from low-tau distributions. This GMM probabilistic value was then multiplied with the tau-PET SUVR to obtain a SUVR score that was cleaned from unspecific signal. To define tau-PET epicenters at the individual level, the top 10 regions with the highest probability of SUVR weighted by GMM were selected and used for further connectivity-based analyses[27,36].

## Functional connectivity template and analyses

To investigate if tau-PET accumulation was linked to functional brain architecture, we derived a template functional connectivity matrix from 69 CU from the ADNI cohort who were Aβ-negative and have low tau-PET binding (global SUVR < 1.3). The steps to derive the template matrix have been described elsewhere[36]. Briefly, functional images were realigned to the first volume and co-registered to the native T1 images. Further processing included detrending, band-pass filtering (0.01–0.08 Hz), and regressing out nuisance covariates (average white matter and CSF signal and motion parameters). Scrubbing of frames with frame displacement greater than 1 mm (along with the frame prior and the two subsequent frames) was applied and only participants with less than 30% of data had to be censored were retained. Functional connectivity matrices were created according to the functional Schaefer atlas of 200 parcels. Fisher-z correlations between time-series averaged across voxels within each ROI were determined to assess subject-specific functional connectivity matrices. All individual matrices were averaged and thresholded at 30% density. The average functional connectivity was then converted to a distance-based connectivity matrix, where then shorter path-length between ROIs represent stronger connectivity[68]. To link functional connectivity and tau aggregates accumulation, we calculated the distance-based functional connectivity of brain regions to the tau epicenters as defined above. Both at the group level and at the individual level, across all remaining brain region not an epicenter ($n = 190$), we correlated tau-PET rate of change in each region to its connectivity-based distance to the tau epicenters. As such, we could measure the strength of the association between tau-PET accumulation and connectivity to tau epicenters across the whole brain, represented as a standardized β-value for each participant. Negative β-values were expected, meaning that stronger connectivity (represented as smaller values given that connectivity measures were converted to distance-based) would be associated with greater tau-PET change. Note that repeating this same approach when adjusting for Euclidean distance between ROIs, or when using a functional connectivity matrix based on partial correlations as template, results were unchanged. From this connectivity-based analyses, for each participant, we then grouped the 190 non-epicenter regions into quartiles based on their connectivity to the tau epicenters (Q1 to Q4). The top 25% of regions with highest connectivity to the tau epicenters were part of Q1, the following 25% in Q2, etc. Average tau-PET rate of change in each quartile was calculated and used in further statistical analyses.

## Statistical analysis

All analyses were performed separately in the non-demented group (CU and MCI Aβ-positive) vs. the AD dementia group in BioFINDER-2. In ADNI analyses were restricted to the non-demented group. Importantly, given the smaller sample size in ADNI, only analyses with sufficient power (i.e., 80% and significance level of 0.05) based on effect sizes from BioFINDER-2 were performed in ADNI. The sample size estimates necessary to conduct each analysis in ADNI was calculated using the R package *pwr* v1.3-0. Details pertaining to the ADNI cohort are presented in the Extended Data. All analyses were performed in R version 4.0.5. The main packages used were *stats* v4.0.5, *lme4* v1.1-30 for linear mixed effect models, *mediation* v4.5.0 for mediation analyses, and *ggplot2* v3.3.6 for creating plots. Brain renderings were created using the Connectome WorkBench software v.1.5.0.

As a first step, we investigated the main factors related to accumulation of tau aggregates (measured as tau-PET rate of change), focusing on regional Aβ-PET SUVR and CSF p-tau217. In each of the 200 brain parcels, we applied different linear regression models, with tau-PET rate of change in each parcel as the dependent variable and age and sex as covariates. Models included progressively more dependent variables, starting with (1) regional Aβ-PET SUVR alone, (2) CSF p-tau217 alone, (3) regional Aβ-PET SUVR and CSF p-tau217, and (4) regional Aβ-PET SUVR, CSF p-tau217 and baseline regional tau-PET SUVR. We report results in regions where coefficients of regional Aβ-PET SUVR or CSF p-tau217 were considered significant, at a p-value <0.05 after FDR correction. We also ensured that there was no multi-collinearity between Aβ-PET SUVR and CSF p-tau217, with variance inflation factors ranging between 1.04 and 1.34. Based on the results from linear models, we also tested the mediating effect of CSF p-tau on regional Aβ-PET SUVR and regional tau-PET rate of change region-wise. Mediation analyses were conducted using the R package *mediation* version 4.5.0. All paths of the mediation model were controlled for age and sex. Significance of the mediation effect was determined using 1000 bootstrapping iterations.

For each participant, linear models were fitted across all non-epicenter regions between connectivity to tau epicenters and tau-PET rate of change, resulting in a β-value. This β-value was then correlated with CSF p-tau concentrations, adjusting for age, sex and global Aβ-PET. Repeated measures ANOVA focusing on the interaction between CSF p-tau concentrations and rate of change in the different quartiles are also reported.

Focusing on the associations between tau measures and cognitive decline, we tested the mediating effect of (1) the β-value linking connectivity and tau aggregates accumulation across the brain and (2) tau-PET rate of change in Q1 on CSF p-tau and cognitive decline (measured either as slope of cognitive composite score or slope of MMSE). Mediation models were performed as mentioned above. All paths of these mediation models were controlled for age, sex and years of education. Correlations between baseline and longitudinal variables are also reported for descriptive purposes in Supplementary Table 2.

### Reporting summary

Further information on research design is available in the Nature Research Reporting Summary linked to this article.

## Data availability

Pseudonymized data from BioFINDER-2 (PI: OH) can be shared to qualified academic researchers after request for the purpose of replicating procedures and results presented in the study. In line with the EU General Data Protection Regulation (GDPR) legislation, a data transfer agreement must be established to share data. The agreement will include terms of how data is stored, protected, and accessed, and will define what the receiver can or cannot do. The agreement also must be approved by the Ethical Review Board of Sweden and Region Skåne. ADNI data used in this manuscript are publicly available from the ADNI database (adni.loni.usc.edu) upon registration and compliance with the data use agreement. Source data are provided with this paper, namely the functional connectivity template and summary statistics. The Schaefer 200 parcels atlas is openly available from the nilearn python package [nilearn.datasets.fetch_atlas_schaefer_2018(n_rois=200)].

## Code availability

Code to perform statistical analyses was written in R and has been made available at https://github.com/alexapichet/NatureComms2022_tau.

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

## Acknowledgements

We would like to acknowledge all the BioFINDER team members as well as participants in the study and their family members for their dedication. Acknowledgement is also made to the donors of the Alzheimer's Disease Research, a program of the BrightFocus Foundation, for support of this research (A2021013F, A.P.B. and O.H.). A.P.B. is supported by a postdoctoral fellowship from the Fonds de recherche en Santé Québec (298314, A.P.B.). Work at the authors' research center was supported by the Swedish Research Council (2016-00906, O.H.), the Knut and Alice Wallenberg foundation (2017-0383, O.H.), the Marianne and Marcus Wallenberg foundation (2015.0125, O.H.), the Strategic Research Area MultiPark (Multidisciplinary Research in Parkinson's disease) at Lund University, the Swedish Alzheimer Foundation (AF-939932, O.H.), the Swedish Brain Foundation (FO2021-0293), The Parkinson foundation of Sweden (1280/20, O.H.), the Konung Gustaf V:s och Drottning Victorias Frimurarestiftelse, the Skåne University Hospital Foundation (2020-O000028, O.H.), Regionalt Forskningsstöd (2020-0314, O.H.) and the Swedish federal government under the ALF agreement (2018-Projekt0279, O.H.). The precursor of $^{18}$F-flutemetamol was sponsored by GE Healthcare. The precursor of $^{18}$F-RO948 was provided by Roche. The funding sources had no role in the design and conduct of the study; in the collection, analysis, interpretation of the data; or in the preparation, review, or approval of the manuscript. Data used in preparation of this article were obtained from the Alzheimer's Disease Neuroimaging Initiative (ADNI) database (adni.loni.usc.edu). As such, the investigators within the ADNI contributed to the design and implementation of ADNI and/or provided data but did not participate in analysis or writing of this report. A complete listing of ADNI investigators can be found at: http://adni.loni.usc.edu/wp-content/uploads/how_to_apply/ADNI_Acknowledgement_List.pdf. Data collection and sharing for this project was funded by the Alzheimer's Disease Neuroimaging Initiative (ADNI) (National Institutes of Health Grant U01 AG024904) and DOD ADNI (Department of Defense award number W81XWH-12-2-0012). ADNI is funded by the National Institute on Aging, the National Institute of Biomedical Imaging and Bioengineering, and through generous contributions from the following: AbbVie, Alzheimer's Association; Alzheimer's Drug Discovery Foundation; Araclon Biotech; BioClinica, Inc.; Biogen; Bristol-Myers Squibb Company; CereSpir, Inc.; Cogstate; Eisai Inc.; Elan Pharmaceuticals, Inc.; Eli Lilly and Company; EuroImmun; F. Hoffmann-La Roche Ltd and its affiliated company Genentech, Inc.; Fujirebio; GE Healthcare; IXICO Ltd.; Janssen Alzheimer Immunotherapy Research & Development, LLC.; Johnson & Johnson Pharmaceutical Research & Development LLC.; Lumosity; Lundbeck; Merck & Co., Inc.; Meso Scale Diagnostics, LLC.; NeuroRx Research; Neurotrack Technologies; Novartis Pharmaceuticals Corporation; Pfizer Inc.; Piramal Imaging; Servier; Takeda Pharmaceutical Company; and Transition Therapeutics. The Canadian Institutes of Health Research is providing funds to support ADNI clinical sites in Canada. Private sector contributions are facilitated by the Foundation for the National Institutes of Health (www.fnih.org). The grantee organization is the Northern California Institute for Research and Education, and the study is coordinated by the Alzheimer's Therapeutic Research Institute at the University of Southern California. ADNI data are disseminated by the Laboratory for Neuro Imaging at the University of Southern California.

## Author contributions

A.P.B., N.F., R.O. and O.H. designed the study. A.P.B. (for BioFINDER-2) and NF (for ADNI) had full access to raw data and carried out the final statistical analyses. A.P.B., N.F., R.O. and O.H. wrote the manuscript and A.P.B. had the final responsibility to submit for publication. All authors (A.P.B., N.F., N.S., M.E., M.B., D.B., O.S., S.J., S.P., N.M.C., R.S., E.S., R.O., O.H.) contributed to the interpretation of the results and critically reviewed the manuscript. The manuscript was also approved by ADNI for publication.

## Funding

## Competing interests

O.H. has acquired research support (for the institution) from AVID Radiopharmaceuticals, Biogen, Eli Lilly, Eisai, Fujirebio, GE Healthcare, Pfizer, and Roche. In the past 2 years, he has received consultancy/speaker fees from Amylyx, Alzpath, Biogen, Cerveau, Fujirebio, Genentech, Roche, and Siemens. S.P. has served on scientific advisory boards and/or given lectures in symposia sponsored by F. Hoffmann-La Roche, Biogen, and Geras Solutions. The remaining others declare no competing interests.

## Additional information

## Alzheimer's Disease Neuroimaging Initiative

Michael Weiner[10], Paul Aisen[11], Ronald Petersen[12], Clifford R. Jack Jr.[12], William Jagust[13], John Q. Trojanowki[14], Arthur W. Toga[15], Laurel Beckett[16], Robert C. Green[17], Andrew J. Saykin[18], John Morris[19], Leslie M. Shaw[20], Enchi Liu[21], Tom Montine[22], Ronald G. Thomas[11], Michael Donohue[11], Sarah Walter[11], Devon Gessert[11], Tamie Sather[11], Gus Jiminez[11], Danielle Harvey[16], Michael Donohue[11], Matthew Bernstein[12], Nick Fox[23], Paul Thompson[24], Norbert Schuff[25], Charles DeCArli[16], Bret Borowski[26], Jeff Gunter[26], Matt Senjem[26], Prashanthi Vemuri[26], David Jones[26], Kejal Kantarci[26], Chad Ward[26], Robert A. Koeppe[27], Norm Foster[28], Eric M. Reiman[29], Kewei Chen[29], Chet Mathis[30], Susan Landau[13], Nigel J. Cairns[19], Erin Householder[19], Lisa Taylor Reinwald[19], Virginia Lee[31], Magdalena Korecka[31], Michal Figurski[31], Karen Crawford[15], Scott Neu[15], Tatiana M. Foroud[18], Steven Potkin[32], Li Shen[18], Faber Kelley[18], Sungeun Kim[18], Kwangsik Nho[18], Zaven Kachaturian[33], Richard Frank[34], Peter J. Snyder[35], Susan Molchan[36], Jeffrey Kaye[37], Joseph Quinn[37], Betty Lind[37], Raina Carter[37], Sara Dolen[37], Lon S. Schneider[38], Sonia Pawluczyk[38], Mauricio Beccera[38], Liberty Teodoro[38], Bryan M. Spann[38], James Brewer[39], Helen Vanderswag[39], Adam Fleisher[39], Judith L. Heidebrink[27], Joanne L. Lord[27], Ronald Petersen[12], Sara S. Mason[12], Colleen S. Albers[12], David Knopman[12], Kris Johnson[12], Rachelle S. Doody[40], Javier Villanueva Meyer[40], Munir Chowdhury[40], Susan Rountree[40], Mimi Dang[40], Yaakov Stern[41], Lawrence S. Honig[41], Karen L. Bell[41], Beau Ances[42], John C. Morris[42], Maria Carroll[42], Sue Leon[42], Erin Householder[42], Mark A. Mintun[42], Stacy Schneider[42], Angela OliverNG[43], Randall Griffith[43], David Clark[43], David Geldmacher[43], John Brockington[43], Erik Roberson[43], Hillel Grossman[44], Effie Mitsis[44], Leyla deToledo-Morrell[45], Raj C. Shah[45], Ranjan Duara[46], Daniel Varon[46], Maria T. Greig[46], Peggy Roberts[46], Marilyn Albert[47], Chiadi Onyike[47], Daniel D'Agostino II[47], Stephanie Kielb[47], James E. Galvin[48], Dana M. Pogorelec[48], Brittany Cerbone[48], Christina A. Michel[48], Henry Rusinek[48], Mony J. de Leon[48], Lidia Glodzik[48], Susan De Santi[48], P. Murali Doraiswamy[49], Jeffrey R. Petrella[49], Terence Z. Wong[49], Steven E. Arnold[20], Jason H. Karlawish[20], David Wolk[20], Charles D. Smith[50], Greg Jicha[50], Peter Hardy[50], Partha Sinha[50], Elizabeth Oates[50], Gary Conrad[50], Oscar L. Lopez[30], MaryAnn Oakley[30], Donna M. Simpson[30], Anton P. Porsteinsson[51], Bonnie S. Goldstein[51], Kim Martin[51], Kelly M. Makino[51], M. Saleem Ismail[51], Connie Brand[51], Ruth A. Mulnard[52], Gaby Thai[52], Catherine Mc Adams Ortiz[52], Kyle Womack[53], Dana Mathews[53], Mary Quiceno[53], Ramon Diaz Arrastia[53], Richard King[53], Myron Weiner[53], Kristen Martin Cook[53], Michael DeVous[53], Allan I. Levey[54], James J. Lah[54], Janet S. Cellar[54], Jeffrey M. Burns[55], Heather S. Anderson[55], Russell H. Swerdlow[55], Liana Apostolova[56], Kathleen Tingus[56], Ellen Woo[56], Daniel H. S. Silverman[56], Po H. Lu[56], George Bartzokis[56], Neill R. Graff Radford[57], Francine ParfittH[57], Tracy Kendall[57], Heather Johnson[57], Martin R. Farlow[18], Ann Marie Hake[18], Brandy R. Matthews[18], Scott Herring[18], Cynthia Hunt[18], Christopher H. van Dyck[58], Richard E. Carson[58], Martha G. MacAvoy[58], Howard Chertkow[59], Howard Bergman[59], Chris Hosein[59], Sandra Black[60], Bojana Stefanovic[60], Curtis Caldwell[60], Ging Yuek Robin Hsiung[61], Howard Feldman[61], Benita Mudge[61], Michele Assaly Past[61], Andrew Kertesz[62], John Rogers[62], Dick Trost[62], Charles Bernick[63], Donna Munic[63], Diana Kerwin[64], Marek Marsel Mesulam[64], Kristine Lipowski[64], Chuang Kuo Wu[64], Nancy Johnson[64], Carl Sadowsky[65], Walter Martinez[65], Teresa Villena[65], Raymond Scott Turner[66], Kathleen Johnson[66], Brigid Reynolds[66], Reisa A. Sperling[67], Keith A. Johnson[67], Gad Marshall[67], Meghan Frey[67], Jerome Yesavage[68], Joy L. Taylor[68], Barton Lane[68], Allyson Rosen[68], Jared Tinklenberg[68], Marwan N. Sabbagh[69], Christine M. Belden[69], Sandra A. Jacobson[69], Sherye A. Sirrel[69], Neil Kowall[70], Ronald Killiany[70], Andrew E. Budson[70], Alexander Norbash[70], Patricia Lynn Johnson[70], Thomas O. Obisesan[71], Saba Wolday[71], Joanne Allard[71], Alan Lerner[72], Paula Ogrocki[72], Leon Hudson[72], Evan Fletcher[73], Owen Carmichael[73], John Olichney[73], Charles DeCarli[73], Smita Kittur[74], Michael Borrie[75], T. Y. Lee[75], Rob Bartha[75], Sterling Johnson[76], Sanjay Asthana[76], Cynthia M. Carlsson[76], Steven G. Potkin[77], Adrian Preda[77], Dana Nguyen[77], Pierre Tariot[29], Adam Fleisher[29], Stephanie Reeder[29], Vernice Bates[78], Horacio Capote[78], Michelle Rainka[78], Douglas W. Scharre[79], Maria Kataki[79], Anahita Adeli[79], Earl A. Zimmerman[80], Dzintra Celmins[80], Alice D. Brown[80], Godfrey D. Pearlson[81], Karen Blank[81], Karen Anderson[81], Robert B. Santulli[82], Tamar J. Kitzmiller[82], Eben S. Schwartz[82], Kaycee M. SinkS[83], Jeff D. Williamson[83], Pradeep Garg[83], Franklin Watkins[83], Brian R. Ott[84], Henry Querfurth[84], Geoffrey Tremont[84], Stephen Salloway[85], Paul Malloy[85], Stephen Correia[85], Howard J. Rosen[10], Bruce L. Miller[10], Jacobo Mintzer[86], Kenneth Spicer[86], David Bachman[86], Elizabether Finger[87], Stephen Pasternak[87], Irina Rachinsky[87], John Rogers[87], Andrew Kertesz[87], Dick Drost[87], Nunzio Pomara[88], Raymundo Hernando[88], Antero Sarrael[88], Susan K. Schultz[89], Laura L. Boles Ponto[89], Hyungsub Shim[89], Karen Elizabeth Smith[89], Norman Relkin[90], Gloria Chaing[90], Lisa Raudin[90], Amanda Smith[91], Kristin Fargher[91] & Balebail Ashok Raj[91]

[10]UC San Francisco, San Francisco, CA, USA. [11]UC San Diego, San Diego, CA, USA. [12]Mayo Clinic, Rochester, NY, USA. [13]UC Berkeley, Berkeley, CA, USA. [14]U Pennsylvania, Philadelphia, PA, USA. [15]USC, Los Angeles, CA, USA. [16]UC Davis, Davis, CA, USA. [17]Brigham and Women's Hospital, Harvard Medical School, Boston, MA, USA. [18]Indiana University, Bloomington, IN, USA. [19]Washington University School of Medicine, St. Louis, MO, USA. [20]University of Pennsylvania, Pennsylvania, CA, USA. [21]Janssen Alzheimer Immunotherapy, South San Francisco, CA, USA. [22]University of Washington, Seattle, WA, USA. [23]University of

London, London, UK. [24]USC School of Medicine, Los Angeles, CA, USA. [25]UCSF MRI, San Francisco, CA, USA. [26]Mayo Clinic, Rochester, NY, USA. [27]University of Michigan, Ann Arbor, MI, USA. [28]University of Utah, Salt Lake City, UT, USA. [29]Banner Alzheimer's Institute, Phoenix, AZ, USA. [30]University of Pittsburgh, Pittsburgh, PA, USA. [31]UPenn School of Medicine, Philadelphia, PA, USA. [32]UC Irvine, Newport Beach, CA, USA. [33]Khachaturian, Radebaugh & Associates, Inc and Alzheimer's Association's Ronald and Nancy Reagan's Research Institute, Potomac, MD, USA. [34]General Electric, Boston, MA, USA. [35]Brown University, Providence, RI, USA. [36]National Institute on Aging/National Institutes of Health, Bethesda, MD, USA. [37]Oregon Health and Science University, Portland, OR, USA. [38]University of Southern California, Los Angeles, CA, USA. [39]University of California San Diego, San Diego, CA, USA. [40]Baylor College of Medicine, Houston, TX, USA. [41]Columbia University Medical Center, New York, NY, USA. [42]Washington University St. Louis, St. Louis, MO, USA. [43]University of Alabama Birmingham, Birmingham, MO, USA. [44]Mount Sinai School of Medicine, New York, NY, USA. [45]Rush University Medical Center, Chicago, IL, USA. [46]Wien Center, Vienna, Austria. [47]Johns Hopkins University, Baltimore, MD, USA. [48]New York University, New York, NY, USA. [49]Duke University Medical Center, Durham, NC, USA. [50]University of Kentucky, Lexington, NC, USA. [51]University of Rochester Medical Center, Rochester, NY, USA. [52]University of California, Irvine, CA, USA. [53]University of Texas Southwestern Medical School, Dallas, TX, USA. [54]Emory University, Atlanta, GA, USA. [55]University of Kansas, Medical Center, Lawrence, KS, USA. [56]University of California, Los Angeles, CA, USA. [57]Mayo Clinic, Jacksonville, FL, USA. [58]Yale University School of Medicine, New Haven, CT, USA. [59]McGill Univ., Montreal Jewish General Hospital, Montreal, QC, Canada. [60]Sunnybrook Health Sciences, Toronto, ON, Canada. [61]UBC Clinic for AD & Related Disorders, Vancouver, BC, Canada. [62]Cognitive Neurology St Joseph's, Toronto, ON, Canada. [63]Cleveland Clinic Lou Ruvo Center for Brain Health, Las Vegas, NV, USA. [64]Northwestern University, Evanston, IL, USA. [65]Premiere Research Inst Palm Beach Neurology, West Palm Beach, FL, USA. [66]Georgetown University Medical Center, Washington, DC, USA. [67]Brigham and Women's Hospital, Harvard Medical School, Boston, MA, USA. [68]Stanford University, Santa Clara County, CA, USA. [69]Banner Sun Health Research Institute, Sun City, AZ, USA. [70]Boston University, Boston, MA, USA. [71]Howard University, Washington, DC, USA. [72]Case Western Reserve University, Cleveland, OH, USA. [73]University of California, Davis Sacramento, CA, USA. [74]Neuro-logical Care of CNY, New York, NY, USA. [75]Parkwood Hospital, Parkwood, CA, USA. [76]University of Wisconsin, Madison, WI, USA. [77]University of California, Irvine BIC, Irvine, CA, USA. [78]Dent Neurologic Institute, Amherst, MA, USA. [79]Ohio State University, Columbus, OH, USA. [80]Albany Medical College, Albany, NY, USA. [81]Hartford Hospital, Olin Neuropsychiatry Research Center, Hartford, CT, USA. [82]Dartmouth Hitchcock Medical Center, Albany, NY, USA. [83]Wake Forest University Health Sciences, Winston-Salem, NC, USA. [84]Rhode Island Hospital, Providence, RI, USA. [85]Butler Hospital, Providence, RI, USA. [86]Medical University South Carolina, Charleston, SC, USA. [87]St. Joseph's Health Care, London, ON, Canada. [88]Nathan Kline Institute, Orangeburg, SC, USA. [89]University of Iowa College of Medicine, Iowa City, IA, USA. [90]Cornell University, Ithaca, NY, USA. [91]University of South Florida, USF Health Byrd Alzheimer's Institute, Tampa, FL, USA.

