## [Peer Review File · Nature Communications]

Amyloid-associated increases in soluble tau relate to tau aggregation rates and cognitive decline in early Alzheimer's diseaseREVIEWER COMMENTS

Reviewer #1 (Remarks to the Author):

Binette et al. investigate the associations between, A β aggregation (A β PET), soluble phosphorylated tau (p-tau217), accumulation of aggregated tau (as assessed by tau PET) and cognitive decline in the AD continuum. They demonstrate that, in early AD (CU + MCI, non-demented), higher CSF p-tau217 was associated with higher Tau PET uptake, higher connectivity-mediated tau accumulation and higher cognitive decline. In mediation analyses, they show that CSF p-tau217 mediates the association between A β and Tau PET uptake, while Tau PET mediates the association between CSF p-tau217 and cognitive decline. In contrast, in the dementia stage, cognitive decline was associated with Tau PET, but not A β or CSF p-tau217.

The major strengths of this study are the use of both cross-sectional and longitudinal data and the multimodal approach (CSF, Tau PET, A β PET, cognitive assessment). That allows the author to propose a model of tau pathology in the AD continuum. Importantly, the authors also validated their main results in the ADNI cohort.

My main concern is why the authors did not include soluble A β , and only A β PET, in these analyses in the prodementia stage. Surprisingly, they did so when studying the dementia stage. If the aim is to study soluble and aggregated tau pathology in early stages, the role of soluble A β is key. Some isoforms of p-tau, particularly p-tau217, increase very early in the continuum. In line with this comment, I would be more cautious in the mechanistic implications of the model they proposed in Figure 6. I suggest to avoid statements like "In early AD, A β triggers increased concentrations and secretion of soluble p-tau". The authors assume that A β aggregates are the trigger of tau cascade, ignoring the role of soluble A β . I understand this is a communication with limited space but, at least, this needs to be explained.

Minor points:

- In the introduction, and if there is space for additional references, I would recommend to include citations from the seminal works in tau spreading (Frost et al., JBC 2009; Clavaguera et al. Nat Cell Biol 2009; de Calignon et al. Neuron 2012).**
- For more clarity, I would recommend to indicate the phosphorylation site of p-tau in the whole text, that is "CSF p-tau217", instead of "CSF p-tau". Many readers would assume that p-tau refers to p-tau181. In fact, the authors use CSF p-tau181 for ADNI.**

Reviewer #2 (Remarks to the Author):

The authors examine associations between baseline Ab-PET and CSF p-tau217 on changes in tau-PET, connectivity-associations with changes in tau-PET and cognitive decline in participants (CU Ab-, CU Ab+, MCI Ab+, Dementia Ab+) from BIOFINDER-2 and replicated, to a degree, in ADNI. They find that CSF p-tau217 is most strongly associated with changes in tau-PET and that the association between CSF p-tau217 and cognitive decline might be mediated by connectivity-associated tau accumulation. While this paper is interesting to read, it does call to mind the vast literature that has been published on this topic in recent years and it does become a little hard to extricate the findings of this work from those that have come before it. The authors mention that this work involves a lengthier follow up with tau-PET, but I am left wondering what the innovative new finding is from this work that can provide new context with which to think about the earliest stages of the AD pathological cascade beyond what we already know from BIOFINDER and ADNI publications. If the authors can provide a stronger argument for this, then my reading of the work will become much more positive. As a strength, the authors should be commended for writing this manuscript clearly and putting forward some well-conducted analyses.

I have further thoughts below:

Considering the premise of this paper hinges upon the method being examined (CSF, plasma or PET), it would be worthwhile being more explicit in the introduction about what citations are referring to what modality. As just one example, the authors refer to p-tau increasing in preclinical AD but then cite both plasma and CSF results, which may deserve a little further unpacking. While CSF and plasma align in some contexts, it is certainly possible that each modality (even within the same 'soluble tau' context) provides slightly different information at different stages about the molecular pathology.

While this study is well conducted, it certainly repeats a similar line of inquiry that has been published fairly prolifically over the past few years and it remains unclear to this reviewer exactly what the novel and innovative step is. The authors argue for the use of more longitudinal data rather than cross-sectional, however, this has arguably been examined in a few cohorts to date. I do see the value in exploring these types of questions, but there may need to be a slightly stronger case made by the authors in the introduction for why this particular paper sets itself apart and how it adds unique information to the field beyond what has been published very recently.

The authors pair the CU and MCI participants but it would make more sense to separate these. The authors would not have enough statistical power to examine MCI, but have they just examined these findings in CU only?

The authors refer to change in tau-PET and cognition in the results, but the reader would benefit to know a few extra details such as the reference region in these longitudinal analyses. The authors mention cerebellar grey matter for longitudinal tau-PET, but this reference region is not very stable over time. Did the authors examine other potential reference regions for these analyses?

In addition, how these slopes were extracted would be ideal to know (linear mixed models or ordinary least squares), along with whether random effects were adjusted for and any other covariates (age, sex, etc). Further, it would be good to state that this is a ROI-wide tau-PET analysis, not using composites. And finally, knowing the time frame over which these slopes are gathered would be critical for the reader to know (i.e. both number of visits and years of collection).

Would be worth mentioning the multiple comparison correction early on.

Was how was partial volume correction calculated? And did the authors conduct analyses with both PVC and non-PVC data (at least those with CU participants)?

On page 5 (line 135) the authors refer to "early on", but what does this mean? And is this a statistical comparison or simply a description of the figures? Would be ideal to refer more specifically to the groupings, such as CU Ab+ vs CU Ab-. The authors then choose a region with the highest accumulation in the dementia stage, was this conducted by rank ordering in the dementia group? Did the authors compare the groups in their rates of change across all 200 parcellated regions? This is not clear. If so, did the authors perform FDR correction or something similar?

Given the large age discrepancy between CU Ab- and non-demented Ab+ did the authors consider trying a sensitivity analysis that age-matches these groups (and potentially even the Ab+ dementia patients, if possible) and then assesses these relationships again? Age is such a murky variable in it's association with pathological change and it becomes very hard to assess these associations when age is such a large confound, as it is here between these groups, particularly considering the CU Ab- are so much younger

The mPACC is so much lower in the AB+ non-demented group that I really suggests to me that the MCI group is having a huge impact on this grouping. Is it at all possible to split these groups up or to just examine the impact of CU AB+ alone? That is, is this

really just the MCIs telling us something (i.e. once cognitive impairment is already a factor)? I suspect that it is, and I feel that this is an important story to acknowledge.

Given the association between Ab-PET and CSF p-tau, at least at the 'earlier' levels of Ab burden, did the authors consider how there might be shared variance explaining these 'separate' relationships with changes in tau-PET burden. That is, these analyses are non-independent and trying to examine the main effects in separate analyses might not be enough to tell us about the independent impacts of Ab-PET and CSF p-tau? Perhaps considerations of PCA or partial least squares (where a factor analysis of Ab and tau to extract shared and separate sources of variance) might help us to get at this question a bit better?

My concern is particularly apparent for the analyses where both Ab and p-tau are included in the model as main effects. The loss of significance of one variable is potentially an artifact of warped estimates due to multicollinearity.

Did the authors ever examine CSF tau217/Ab42 ratio instead of just CSF217? I am curious as to whether this might provide stronger associations or not.

Many of the Ab-PET ROIs are also non-independent either - that is, they are highly associated with one another and determining whether one region is truly more associated with changes in tau becomes quite difficult. Did the authors consider similar approaches to examining shared variance across regions?

Can the authors report the correlation between p-tau217 and baseline tau-PET?

If one swapped out p-tau217 for baseline tau-PET in figure 3C, would there be more or less explained variance (i.e. R²)?

While there is an association found in Figures 2A and 2B, the findings are quite weak and involve a lot of heterogeneity that I think is worth discussing.

In Figures 4A-D could the authors color the scatterplot according to diagnostic/Ab group?

Could the authors examine whether the tau-PET Q1 rate of change mediation model (Fig 4G) performs any differently relative to the general metric of tau-PET rate of change (i.e. removing the connectivity component)? Or any of the other quartiles? I just want to ascertain how well these models perform relative more comparable mediation variables that are likely to explain the association between CSF p-tau and mPACC change.

If the authors remove the MCI patients, do the connectivity and cognitive findings hold up (i.e. entirely CU cohort)? The inclusion of the MCI participants will introduce extra variance to play with in these models, which is good, but creates a much easier path to finding significance when really one might be simply tapping into those who are already well on their way to a clinical diagnosis of dementia (i.e. very far from preclinical AD or the 'early stages of AD').

The authors mention soluble levels of CSF Ab42/40 in their dementia section but it is not analyzed or mentioned in the previous sections - why is this?

The authors mention ADNI early in their results section but I do not see mention of it again in the results? I may be missing something but it would be worth focusing some sentences on validation for each analysis throughout the section.

The authors mention early in their discussion that ADNI "fully" replicated their results, but there should be caveats here to remind the reader that ADNI is a smaller sample size and that only certain analyses were able to be validated in this cohort due to power issues. Further, this replication really isn't mentioned at all in the results and needs to

be explicitly described. The authors allude to the shortcomings in the limitations section, but needs to be amplified more throughout the text.

Reviewer #3 (Remarks to the Author):

Binette et al. report longitudinal data from the BIOFINDER2 cohort. The PET and fMRI data are based on a brain parcellation atlas. fMRI connectivity is analysed using correlation analysis. CSF 217phosphotau is a predictor of connectivity-dependent increase in tau PET signal and has a stronger effect than regional amyloid PET values. This multimodal longitudinal dataset is very rich in content and relatively unique. The analyses are thorough and well developed. The main findings are based on state-of-the-art statistics (incl. mediation modeling). The findings are novel and of clinical and fundamental relevance and are presented with sufficient detail. My main concern relates to the way in which amyloid load is parameterized throughout the paper.

Major comments

1. The comparison between 217phosphotau as predictor versus regional A β is not entirely symmetrical. There are as many regional A β values as regions examined ($n = 200$), compared to a single CSF parameter. For each of the analyses, the authors should verify whether results remain if A β load is expressed as a single measure based on a composite region. A composite amyloid PET value may be more robust than a set of 200 regional values. This could be the typical composite region used for amyloid imaging or, in the asymptomatic stage, precuneus levels. It is conceivable that A β measures after parcellation into 200 regions yields a substantial portion of regional levels below the cut-off and hence reflects mostly noise. That would inevitably lead to poor results for the A β predictor compared to phosphotau. Furthermore, the regional tau accumulation is correlated with the amyloid load in the same region. The underlying assumption that amyloid accumulation and tau accumulation colocalize, is unlikely.
2. Data from cognitively normal participants and MCI are pooled. There is a strong interest in the earliest stages of AD, when participants are still cognitively normal. Hence it would be critical to verify which results remain when only the cognitively normal participants are examined. Given the relatively slow rate of change of tau load in the cognitively normal stage, it is conceivable that the relations described are no longer significant. That would have a profound repercussion on the interpretation as that would change the paper from a study of the earliest pathophysiological changes in AD to a study of changes occurring during the clinical stages of AD. This of course would remain of interest but the neurobiological interpretation would become different. The designation 'non-demented participants' may be technically correct but from a clinical perspective, however the distinction between a cognitively normal, community-recruited participant versus a memory clinic recruited patient with mild cognitive deficits is crucial. The grouping as 'non-demented' is meant to increase sensitivity by including symptomatic cases who are known to have higher biomarker levels that change more rapidly but is scientifically contentious. Pooling MCI and cognitively normal makes it difficult for the reader to determine whether the findings pertain in the first instance to the asymptomatic AD stage or to the MCI stage.
3. The authors push the interpretation of their data in terms of the underlying neurobiological processes and the pathogenesis of Alzheimer disease. This is laudable and exciting and ideally should be possible but with the current strength of evidence for these biomarkers, the authors could be more cautious about the interpretation of biomarkers in terms of underlying neurobiological processes and early AD pathogenesis. Among their four biomarkers, only amyloid PET is neuropathologically validated by an end-of-life validation data. Amyloid PET mostly detects amyloid stage 4 and 5 with high accuracy and amyloid stage 3 with moderate accuracy. Hence, the amyloid measure does not detect the earliest stages as one may think based on the neurobiological interpretation the authors assign to their in vivo measures. For the other biomarkers, neuropathological validation is missing. For the tau PET tracer, it is plausible that it detects only Braak stage 4 and 5 reliably by analogy with flortaucipir but we (at least the reviewer) do not know for sure. For soluble 217phosphotau there are no end-of-life data that validate this CSF measure as a reliable indicator of a specific neurobiological

process in the brain. In clinical practice we assume that it is AD specific if a stringent cut-off is used, but the meaning of CSF phosphotau as a continuous measure of the level of a specific compound in the brain is not proven strictly speaking. This is a matter of opinion, there is no strong objective evidence to the best of my knowledge and it would be hard to collect such evidence. If there is strong evidence of a continuous relationship between CSF 217phosphotau and brain levels of soluble hyperphosphorylated tau, please cite it in the paper. For the fMRI connectomics the evidence and technical validation in terms of neuroanatomical brain connectivity is also limited to the best of our knowledge, while such evidence would be essential for an interpretation in terms of transsynaptic spread. Again, please cite the relevant evidence demonstrating the validity of the brain fMRI interregional correlation as a measure of mono- or multisynaptic anatomical connectivity in the paper.

Minor comments

4. Like other groups, the authors describe an acceleration of tau PET tracer signal increase with disease stage based on linear change in the asymptomatic, the MCI and the early dementia stage. Across the entire cohort is the increase best captured by an exponential model? If the rate constant of the exponential model is used as outcome measure rather than the linear change, is there still an effect of the same baseline variables? Or is the effect of the baseline variables entirely attributable to where the case is situated on the exponential curve?
5. If baseline tau PET load is included as one of the predictors in the mediation model (Fig 4E-G), does the effect of phosphotau remain or does baseline tau PET load as predictor wipe out all the other effects?
6. The term 'modified PACC' is misleading as the set of cognitive tests differs too much from the PACC. I realize this term has been used by this group in previous study but it should be re-named in a more objective manner rather than suggesting a high similarity with the PACC.
7. Overall the average interval of follow-up remains relatively modest (1.5 yrs), especially for the asymptomatic phase that extends over many years, with a clinical phase that can be 10 years or longer.
8. The fMRI connectivity matrix is based on correlations between time series. This is used as a measure of transsynaptic spread. Logically, partial correlations would be more suited as correlations can be direct or indirect and for transsynaptic spread the distinction between mono- and multi-synaptic connectivity is relevant. Second, correlations between timeseries can be heavily influenced by Euclidean distance. Please explain how you deal with this potential confound.
9. P 26: the authors compare models with different sets of predictors. Please explain how you compare the difference in fit between models and when you decide that the fit differs significantly.
10. Please add the correlation matrix between the baseline variables. The paper is presented as separating out effects of different baseline variables but if these are highly correlated, the distinctions may be less clear-cut. It is also relevant for the mediation model.
11. Fig 4 panel A-D: please clarify whether the association is based on Pearson or Spearman correlation. Given the distribution of the data, this certainly should be based on Spearman correlations.

Very minor comment

12. When change in tau PET is modelled, the authors use 'time in years' as independent variable. The time resolution should be clarified. Ideally one would use scan date and model change over time in that way I guess.
13. To define tau PET centers the authors used a mixed Gaussian model based on the data they have 'since the right-most distribution likely reflects abnormal tau PET signal' (p 24-25). Using an independent dataset to define thresholds of tau pathology per region would be a good alternative.

I hope these comments are helpful.

Sincerely,

Rik Vandenberghe

Reviewer #1:

Pichet Binette et al. investigate the associations between A β aggregation (A β PET), soluble phosphorylated tau (p-tau217), accumulation of aggregated tau (as assessed by tau PET) and cognitive decline in the AD continuum. They demonstrate that, in early AD (CU + MCI, non-demented), higher CSF p-tau217 was associated with higher Tau PET uptake, higher connectivity-mediated tau accumulation and higher cognitive decline. In mediation analyses, they show that CSF p-tau217 mediates the association between A β and Tau PET uptake, while Tau PET mediates the association between CSF p-tau217 and cognitive decline. In contrast, in the dementia stage, cognitive decline was associated with Tau PET, but not A β or CSF p-tau217.

The major strengths of this study are the use of both cross-sectional and longitudinal data and the multimodal approach (CSF, Tau PET, A β PET, cognitive assessment). That allows the author to propose a model of tau pathology in the AD continuum. Importantly, the authors also validated their main results in the ADNI cohort.

1. My main concern is why the authors did not include soluble A β , and only A β PET, in these analyses in the predementia stage. Surprisingly, they did so when studying the dementia stage. If the aim is to study soluble and aggregated tau pathology in early stages, the role of soluble A β is key. Some isoforms of p-tau, particularly p-tau217, increase very early in the continuum. In line with this comment, I would be more cautious in the mechanistic implications of the model they proposed in Figure 6. I suggest to avoid statements like “In early AD, A β triggers increased concentrations and secretion of soluble p-tau”. The authors assume that A β aggregates are the trigger of tau cascade, ignoring the role of soluble A β . I understand this is a communication with limited space but, at least, this needs to be explained.

Authors’ response: We thank the reviewer for the positive assessment of our manuscript, and we agree that we should have considered soluble A β measures to get a fuller picture, especially at the early stages of AD pathogenesis. As such, we have now repeated analyses from Figure 2 on the association between A β and tau accumulation, substituting regional A β PET for CSF A β 42/40. The results remained unchanged when using CSF A β 42/40 instead of amyloid-PET; including both CSF A β 42/40 and p-tau as predictors of tau-PET rate of change, only p-tau217 remains significant (Supplementary Fig. 5B copied below). When looking only at the effect of the CSF A β 42/40 ratio, the CSF ratio is related to tau-PET rate of change in fewer similarly compared to regional A β . We should also point out that we used state-of-the-art CSF assays, both the Elecsys A β 42/40 ratio and MSD p-tau217 have shown very high AUC to identify AD pathology. CSF A β 42 was also available in ADNI, and repeating similar analyses in this cohort also revealed similar results as when using A β -PET (Supplementary Fig. 7C copied below)

We now mention these results in the main Results section (page 9): “Results were also consistent across different measures of A β , i.e. if using global A β load assessed with PET or the CSF A β 42/40 ratio instead of regional A β aggregates in regression models (Supplementary Fig. 5).”

We also modified our claim in the legend of Figure 6, to include CSF A β , given that our results were found both with A β measures obtained from CSF and PET. The sentence now reads “In early AD, **soluble or insoluble** A β triggers increased concentrations and secretion of soluble p-tau” (page 21).

Supplementary Fig. 5. CSF p-tau217 and A β associations using different measures of A β with regional tau-PET [18 F]RO948 rate of change in A β -positive non-demented participants

Similar analyses as done on Fig. 2, using different amyloid measure instead of regional A β -PET. The amyloid measure is global A β -PET SUVR in **A** and CSF A β 42/40 ratio in **B**. In both panels, the left two columns show standardized beta coefficient in regions where A β and CSF p-tau respectively relates to regional tau-PET rate of change, adjusting for age and sex. Two middle columns show standardized beta coefficient where A β and CSF p-tau217 are associated to regional tau-PET rate of change when including both biomarkers in the same model, adjusting for age and sex (tau PET rate of change \sim A β + CSF p-tau217 + age + sex). Right two columns show the same depiction as in middle columns when additionally controlling for regional baseline tau-PET SUVR. All regions shown on the brain are significant at p<0.05 after FDR-correction.

Supplementary Fig. 7. CSF p-tau181 and A β associations with regional tau aggregates accumulation in A β -positive non-demented participants in ADNI

(A) Standardized beta coefficient of CSF p-tau181 in regions where CSF p-tau181 relates to regional tau-PET rate of change, adjusting for age and sex (left). Standardized beta coefficient of CSF p-tau181 in regions where CSF p-tau181 relates to regional tau-PET rate of change, when adjusting for global A β centiloid value, age and sex (middle), and when also adjusting for regional baseline tau-PET SUVR (right). **(B)** Mediating effect of CSF p-tau181 on global A β centiloid value accumulation of tau aggregates across the brain (average rate of change across 200 regions). **(C)** Same analyses as shown in A, but when using CSF A β 42 as the amyloid measure. All regions shown on the brain are significant at $p < 0.05$ after FDR-correction.

Minor points:

- In the introduction, and if there is space for additional references, I would recommend to include citations from the seminal works in tau spreading (Frost et al., JBC 2009; Clavaguera et al. Nat Cel Biol 2009; de Calignon et al. Neuron 2012).

Authors' response: Thank you for pointing out these important papers. We have now added two of the suggested references in the introduction (de Calignon et al, 2012, Frost et al, 2009; References 19 and 26).

- For more clarity, I would recommend to indicate the phosphorylation site of p-tau in the whole text, that is "CSF p-tau217", instead of "CSF p-tau". Many readers would assume that p-tau refers to p-tau181. In fact, the authors use CSF p-tau181 for ADNI.

Authors' response: We have clarified that throughout the manuscript. When we refer to CSF, we have clarified that it is CSF p-tau217 in BioFINDER-2 and CSF p-tau181 in ADNI.

Reviewer #2:

1. The authors examine associations between baseline Ab-PET and CSF p-tau217 on changes in tau-PET, connectivity-associations with changes in tau-PET and cognitive decline in participants (CU Ab-, CU Ab+, MCI Ab+, Dementia Ab+) from BIOFINDER-2 and replicated, to a degree, in ADNI. They find that CSF p-tau217 is most strongly associated with changes in tau-PET and that the association between CSF p-tau217 and cognitive decline might be mediated by connectivity-associated tau accumulation. While this paper is interesting to read, it does call to mind the vast literature that has been published on this topic in recent years and it does become a little hard to extricate the findings of this work from those that have come before it. The authors mention that this work involves a lengthier follow up with tau-PET, but I am left wondering what the innovative new finding is from this work that can provide new context with which to think about the earliest stages of the AD pathological cascade beyond what we already know from BIOFINDER and ADNI publications. If the authors can provide a stronger argument for this, then my reading of the work will become much more positive. As a strength, the authors should be commended for writing this manuscript clearly and putting forward some well-conducted analyses.

Authors' response: We thank the reviewer for the thorough assessment of our manuscript. In addressing all points below, we consolidated the robustness of our main results (e.g. using different amyloid measures such as global amyloid-PET or CSF A β , splitting CU and MCI which strengthens that the pathophysiological processes we are proposing are also at play in the preclinical stage).

Also, we would like to further highlight the major novelty of our study. While we agree with the reviewer that some of the analyses included in our paper – taken individually – are in line with previous papers, the key innovation of our study is to put the individual pieces together and build a full biomarker-based pathophysiological cascade model, that links A β deposition, soluble p-tau levels, tau spreading and aggregation with cognitive decline in the entire AD continuum. Specifically, we conclude that Ab drives earliest p-tau increases, which may in turn be the initial vehicle for trans-neuronal tau spreading, resulting in tau aggregation and cognitive deterioration. Our work is mechanistically motivated and provides *in vivo* evidence for the complex interplay of different pathological processes which may result in the development of AD. In the revision we have now also shown for the first time that A β -induced p-tau increases even seem to drive the accumulation over time of insoluble tau aggregates in cognitively normal individuals. We believe that our work has clear clinical implications, arguing that attenuating p-tau increases in early disease stages (e.g. using antisense oligonucleotides directed against tau production), especially during the preclinical stage of AD, may be a key approach to halt downstream pathological processes that result in the development of cognitive decline. To better point out the novelty of our comprehensive analysis approach, we have modified the introduction of our paper (see answer to point #3). Together, we believe the novelty of the paper now better comes across and have tried to be more explicit about the innovative aspects of the current work.

I have further thoughts below:

2. Considering the premise of this paper hinges upon the method being examined (CSF, plasma or PET), it would be worthwhile being more explicit in the introduction about what citations are referring to what modality. As just one example, the authors refer to p-tau increasing in preclinical AD but then cite both plasma and CSF results, which may deserve a little further unpacking. While CSF and plasma align in some contexts, it is certainly possible that each

modality (even within the same ‘soluble tau’ context) provides slightly different information at different stages about the molecular pathology.

Authors’ response: We appreciate the reviewer’s comment and have clarified this point. Since this paper focuses on CSF only and given the limitation of overall references, we now only reference studies related to CSF p-tau in the first paragraph of the introduction (page 3).

3. While this study is well conducted, it certainly repeats a similar line of inquiry that has been published fairly prolifically over the past few years and it remains unclear to this reviewer exactly what the novel and innovative step is. The authors argue for the use of more longitudinal data rather than cross-sectional, however, this has arguably been examined in a few cohorts to date. I do see the value in exploring these types of questions, but there may need to be a slightly stronger case made by the authors in the introduction for why this particular paper sets itself apart and how it adds unique information to the field beyond what has been published very recently.

Authors’ response: We agree with the reviewer that there are already several papers on the association between soluble p-tau (i.e. either in CSF or plasma) and tau aggregates, to which our group also contributed. Still, we would like to point out that almost all of this literature up to now comes from cross-sectional tau-PET measurements, and very few (if not any) are combining A β and soluble p-tau to better understand the evolution of tau aggregation across time. One paper did investigate the link between soluble p-tau181 and longitudinal tau-PET in ADNI (Moscoso et al, *Alzheimer’s Dementia*, 2022; published after our initial submission), with the focus on optimizing screening for clinical trials using soluble p-tau markers, which is considerably different from our current paper. The only minor overlap in the analyses done between this paper and ours is that we both report that CSF p-tau is related to tau-PET rate of change in ADNI. In contrast, here we investigate how both Ab, p-tau and baseline tau-PET contribute to subsequent tau aggregation, we further link p-tau with connectivity-based tau spreading and then combine different tau measures to evaluate their effect on longitudinal cognitive change in the different stages of AD. As pointed out in our response to point #1, our major goal was to test a complex biological model of AD development, rather than showing mere associations between different biomarkers. Together, the key findings of our paper were that i) Ab-related p-tau increases are a prerequisite for subsequent tau aggregation in early-stage AD, ii) p-tau increases are specifically associated with the spreading of tau across interconnected brain regions, and iii) tau aggregation rates mediate the association between high soluble p-tau and subsequent cognitive decline.

Importantly, we also show that p-tau is a driving force of tau spread and aggregation in early non-dementia stages of AD (both in CU and MCI separately; see response to point #4), while tau aggregation becomes a self-promoting process that is uncoupled from soluble p-tau in late AD stages. To better point out the novelty of our study, we have tightened the introduction and tried to better articulate the unique points of the paper, the key new parts are copied below.

(page 3) “Therefore, increases in soluble p-tau may drive subsequent tau aggregation and spread in AD, suggesting p-tau as a potential treatment target to halt tau spread and the formation of toxic insoluble tau aggregates. However, the cross-link between A β , p-tau and tau aggregation is yet to be better understood in humans across the spectrum of AD.”

(page 4) “Yet, evidence in human studies overwhelmingly stem from cross-sectional studies, and a biologically plausible cascade model that incorporates these individual findings and can

link soluble p-tau with downstream tau aggregation and clinical manifestations across the entire AD spectrum is still missing. Testing this model is critical to clarify the potential of p-tau as a target to attenuate tau aggregation and spread, and the optimal target populations that may benefit from such treatments. This is particularly important since drugs efficiently lowering the soluble tau levels have recently advanced to phase 2 (Clinical Trial ID: NCT05399888). Based on progress in the anti-tau drug development pipeline, it is essential to establish during which phases of the disease soluble p-tau levels are most associated with subsequent accumulation of insoluble tau aggregates and cognitive decline.”

(page 4) “Overall, our study suggests that A β -related increased soluble levels of p-tau is a key driver in the accumulation of tau aggregates and connectivity-mediated tau spreading in early-stage preclinical and prodromal AD, while the accumulation rate of tau aggregates becomes self-promoting in late-stage AD dementia and might no longer be driven by p-tau. We thus suggest that soluble p-tau levels may be an optimal treatment target for attenuating tau aggregation and subsequent cognitive decline in the earliest stages of AD, prior to the development of widespread insoluble fibrillar tau aggregates and dementia, which will be critical for the design of future clinical trials.”

4. The authors pair the CU and MCI participants but it would make more sense to separate these. The authors would not have enough statistical power to examine MCI, but have they just examined these findings in CU only?

Authors’ response: A similar point was raised by Reviewer 3, and we have conducted all main analyses from Figures 2 to 4 when separating the CU and MCI participants. To improve statistical power and strengthen the overall study, we updated our sample from the BioFINDER-2 study to include the latest follow-up PET scans and the latest cognitive assessments. As such, we now have 65 CU and 65 MCI in the non-demented group. Our follow-time was also increased, for an average of 2 years of longitudinal tau-PET and 3 years of longitudinal cognition.

All main results hold when analyzing CU and MCI separately, with the small exception of one mediation analysis linking connectivity-based beta value related to tau-PET change and cognitive decline, which was only found in the whole sample of non-demented Ab-positive participants. For each analysis, we now provide results in each group separately in Supplementary figures and added corresponding information in the main Results, all copied below.

We believe that the replication of our findings in CU patients with elevated A β (i.e. preclinical AD) is a key novelty, showing that soluble p-tau may drive also the earliest aggregation of tau, hence targeting soluble levels of p-tau may be a suitable approach for preventing the development of tau aggregation and ensuing cognitive impairment, especially if initiated during preclinical stages. This finding is of particular clinical importance given that several antisense oligonucleotides directed against tau are now entering phase 1 or phase 2 trials, but all of these target AD patients with cognitive impairment.

Text added in the Results section of the main manuscript:

Further, investigating the same relations in the CU and MCI groups separately, the strong effect of soluble p-tau on subsequent tau aggregation rate, above the effect of A β , was clearly found in both groups (Supplementary Fig. 6). When further adjusting for local baseline tau

aggregates, soluble also remained a significant predictor in temporo-parietal regions in each group (Supplementary Fig. 6). (page 9)

The same associations were found in CU and MCI participants alone (Supplementary Fig. 8), suggesting that this effect is not only due to the MCI participants who tend to show greater association between connectivity to epicenter and tau-PET rate of change (more negative β -values on Fig 3B). (page 12)

The same analyses were also repeated splitting the non-demented group into A β -positive CU and MCI separately. We observe a full mediation effect of tau-PET rate of change, either in Q1 or in the temporal meta-ROI between soluble p-tau and cognitive decline on the cognitive composite score in CU ($c'-c = -0.21 [-0.40, -0.06]$, $p=0.002$), and on decline on MMSE in MCI ($c'-c = -0.24 [-0.46, -0.04]$, $p=0.01$), see Supplementary Fig. 11 for detailed statistics. The only analysis that we could not replicate in individual groups was the mediating effect of the β -value of connectivity-based tau-PET change (Fig. 4F), which was also of smaller magnitude in the full group of non-demented participants. (page 14)

Supplementary Fig. 6. Regional A β -PET and CSF p-tau217 associations with regional tau-PET [18 F]RO948 rate of change in A β -positive CU and MCI separately

Same analyses as reported in Fig. 2 in the full sample of non-demented A β -positive participants, when splitting the group into CU only (**A**) and MCI participants only (**B**). In both panels, the left two columns show standardized beta coefficient in regions where regional A β -PET and CSF p-tau217 respectively relates to regional tau-PET rate of change, adjusting for age and sex. Two middle columns show standardized beta coefficient where regional A β -PET and CSF p-tau217 are associated to regional tau-PET rate of change when including both biomarkers in the same model, adjusting for age and sex (tau PET rate of change \sim regional A β -PET + CSF p-tau217 + age + sex). Right two columns show the same depiction as in middle columns when additionally controlling for regional baseline tau-PET SUVR. All regions shown on the brain are significant at $p < 0.05$ after FDR-correction.

Supplementary Fig. 8. Individualized connectivity-based associations of tau-PET rate of and CSF p-tau217 in CU and MCI Aβ-positive participants separately

Same analyses as on Fig. 3A-C in the full sample of non-demented Aβ-positive participants, done here in CU Aβ-positive only (**A**) and MCI Aβ-positive (**B**). All analyses were significant in CU and MCI separately. In both panels, left scatter plot depicts group-level analysis showing how connectivity to the tau epicenters relates to tau-PET rate of change across the whole brain. Each dot represents a brain region. Regions more strongly functionally connected to the epicenters have greater rate of tau-PET accumulation. Repeating the same approach depicted at the individual level, we could generate for each participant a β-value from the correlation between tau-PET rate of change and connectivity-based distance to epicenters across all brain regions, as shown on the box plot in the middle. Right scatter plots show the association between CSF p-tau217 and the β-values of epicenter connectivity to tau-PET rate of change. Each dot represents an individual. The expected negative association suggests that higher CSF p-tau217 is associated with the overall pattern of tau-PET change in more functionally connected regions to epicenters.

A CU Aβ+
Similar results with:

- tau-PET rate of change in the temporal ROI as mediator ($c' - c = -0.33, p < 0.001$)
- MMSE rate of change as cognitive outcome ($c' - c = -0.26, p < 0.001$)

B MCI Aβ+
Similar results with:

- tau-PET rate of change in the temporal ROI as mediator ($c' - c = -0.26, p = 0.006$)
- No mediation effect with cognitive composite score rate of change as cognitive outcome

Supplementary Fig. 11. Cognitive decline analyses in Aβ-positive CU and MCI separately

Same mediation analyses as in Fig. 4 in the full sample of non-demented Aβ-positive participants, done here in CU Aβ-positive only (**A**) and MCI Aβ-positive (**B**). Main analyses were done using tau-PET rate of change in Q1, as shown on the graphs, but also hold when instead using tau-PET rate of change in the temporal meta-ROI. In CU, results were found with rate of change on the cognitive composite score or MMSE. In MCI, results were only found with MMSE rate of change.

5. The authors refer to change in tau-PET and cognition in the results, but the reader would benefit to know a few extra details such as the reference region in these longitudinal analyses. The authors mention cerebellar grey matter for longitudinal tau-PET, but this reference region is not very stable over time. Did the authors examine other potential reference regions for these analyses?

Authors' response: The reference region that was used for all tau-PET analyses is the inferior cerebellar grey matter, as commonly used (Baker et al, 2017). Given the consensus that this is the preferred reference region across all main groups working with tau-PET, we did not examine another reference region. Please also note that we have previously assessed different reference ROIs (i.e. inferior cerebellar grey vs. eroded white matter) to assess longitudinal tau changes, where we obtained consistent results across both reference regions (Franzmeier et al., *Nat Comms*, 2020)

We clarified this point in the Methods section on Image Acquisition and Processing: “Standardized uptake value ratio (SUVr) images were created using the inferior cerebellar gray matter as the reference region for [18F]RO948” for BioFINDER-2 and “SUVr images were created using the inferior cerebellum as the reference region for flortaucipir” for ADNI (page 24)

6. In addition, how these slopes were extracted would be ideal to know (linear mixed models or ordinary least squares), along with whether random effects were adjusted for and any other covariates (age, sex, etc). Further, it would be good to state that this is a ROI-wide tau-PET analysis, not using composites. And finally, knowing the time frame over which these slopes are gathered would be critical for the reader to know (i.e. both number of visits and years of collection).

Authors' response: The slopes were extracted from linear mixed effect models with random slope and intercept. At this point we were only interested on the effect of time and did not include other covariates. Please note that other covariates such as age, sex etc. were included in second-level analyses, using tau-PET rate of change as a dependent variable.

We added this information in the Methods section Regional measures and rate of change of PET SUVR: *“To calculate the rate of change in tau-PET over time, linear mixed effect models with random slope and intercept were fitted for each brain region, with tau-PET SUVR as the dependent variable and time in years from baseline as the independent variable. (page 26)*

In the main Results, we also state more clearly that the analyses are ROI-based and that the rate of change comes from linear mixed models: *“Accumulation of tau aggregates was quantified as the rate of change in tau-PET retention over time (SUVR/year) in each brain region separately, derived from linear mixed effect models.”(page 5)*

We also added the number of visits, and length of follow-up in Table 1 (page 7).

	Aβ-negative controls (n=204)	Aβ-positive non-demented (n=130)	Aβ-positive AD dementia (n=66)
Age (years)	62.4 ± 15.0	71.7 ± 8.4	72.7 ± 7.3
Sex F (%F)	100 (49%)	63 (49%)	37 (56%)
Education (years)	12.7 ± 3.2	12.4 ± 4.4	11.8 ± 4.3
APOEε4 carriers (%)	73 (36%)	95 (73%)	44 (67%)
CSF p-tau217 (pg/ml)	47.6 ± 31.4	243.7 ± 174.5	570.8 ± 300.9
MMSE	29.0 ± 1.2	27.7 ± 1.9	20.1 ± 4.4
Cognitive composite score	0.01± 0.7	-1.4 ± 1.2	-4.4 ± 1.5
tau-PET follow-up time (years)	2.0 ± 0.6	2.0 ± 0.6	1.7 ± 0.3
Number of PET scans, n (%)			
2	173	92	48
3	29	32	18
4	2	6	0
Cognitive follow-up time (years)	2.4 ± 0.8	3.1 ± 0.7	2.0 ± 0.6
Number of cognitive assessments, n (%)			
2	144	13	11
3	42	52	22
4	12	46	33
5	4	19	0

Table 1. BioFINDER-2 cohort demographics

Data are presented as mean ± standard deviation unless specified otherwise. One Aβ-positive non-demented participant had missing APOE genotype. Cognitive composite are z-scores.

Abbreviations: Aβ= beta-amyloid; APOEε4= apolipoprotein E genotype (carrying at least one ε4 allele); CSF p-tau217= cerebrospinal fluid phosphorylated tau 217; MMSE= Mini-Mental State Examination; PET= positron emission tomography. See Table S1 for ADNI participants.

7. Would be worth mentioning the multiple comparison correction early on.

Authors' response: We now mention it at the beginning of the Results section: “*For analyses done across the 200 brain regions, only results surviving multiple comparisons from false discovery rate are reported.*” (page 5)

8. Was how was partial volume correction calculated? And did the authors conduct analyses with both PVC and non-PVC data (at least those with CU participants)?

Authors' response: The main analyses were conducted on non-PVC data. Up to date, there is no consensus on how crucial PVC is for tau-PET and previous analyses from our group found little difference when using PVC or non-PVC data (Leuzy et al, *JAMA Neurology*, 2022; Franzmeier et al, *Science Advances*, 2020), which motivated our choice to use non-PVC data.

Nevertheless, we confirmed that the rate of change in tau-PET was similar using PVC data (Geometric Transfer Matrix) and that the main effect of CSF p-tau was still found across the brain. We have created the equivalent of Fig. 1 and 2 using PVC-corrected data (figure copied below), which shows results highly consistent with those in the main manuscript. Given the already extensive new analyses added to the study, we did not include these additional analyses in the manuscript but restrict it to the rebuttal. Please note that the rebuttal will be made available online in case the manuscript is accepted for publication, so these confirmatory results will be available for the interested reader.

Analyses using PVC-corrected data

Aβ-positive non-demented group (CU and MCI)

Equivalent of Fig. 1 and 2 from the main manuscript, here using PVC-corrected SUVR as input to calculate tau-PET rate of change.

9. On page 5 (line 135) the authors refer to “early on”, but what does this mean? And is this a statistical comparison or simply a description of the figures? Would be ideal to refer more specifically to the groupings, such as CU Ab+ vs CU Ab-. The authors then choose a region with the highest accumulation in the dementia stage, was this conducted by rank ordering in the dementia group? Did the authors compare the groups in their rates of change across all 200 parcellated regions? This is not clear. If so, did the authors perform FDR correction or something similar?

Authors' response: We changed this section to be more precise and removed the part about the dementia stage, which felt superfluous. Our goal was to report the rate of change in key tau regions, and not to statistically compare groups to one another per se. We find these number important to report to help compare the rate of change in tau-PET in our study with rate of change in other cohorts.

We also applied the reviewer suggestion to do statistical comparisons as an additional description of rate of change between groups. We report maps of group-wise comparisons in Supplementary Fig. 2 (copied below, showing regional p-values surviving post-hoc t-tests comparing regional tau-PET rate of change between groups) and refer to it in the results section.

It now reads: *“In the BioFINDER-2 cohort, the baseline distribution of insoluble tau aggregates assessed via tau-PET recapitulated the AD-typical deposition in the medial and lateral temporal lobes in the controls and non-demented participants, before gradually extending mainly into lateral and medial parietal and lateral occipital regions at symptomatic AD stages (Fig. 1A and Supplementary Fig. 1 for CU and MCI separately) [...] Quantitatively, focusing on regions corresponding to a temporal meta-ROI encompassing medial and lateral parts of the temporal lobe - key regions of tau aggregates early in the disease process and approximating Braak stages I to IV -, the average annual SUVR rate of change was 0.7% in A β -negative controls, 3.3% in A β -positive non-demented individuals, and 9.0% in A β -positive patients with AD dementia. Between-group comparisons showing regions where accumulation of tau aggregates differ are also reported in Supplementary Fig. 2 for complementary description of regional differences of tau-PET rate of change.” (page 5)*

Supplementary Fig. 2. Between group comparisons of tau-PET [¹⁸F]RO948 SUVR rate of change Regions where tau-PET rate of change is significantly different between (A) A β -negative controls vs. A β -positive non demented participants, and (B) A β -positive non demented participants vs. A β -positive AD dementia patients. T-tests were done to compare groups and regions with significant p-value from post-hoc Tukey HSD are reported on the brains.

10. Given the large age discrepancy between CU Ab- and non-demented Ab+ did the authors consider trying a sensitivity analysis that age-matches these groups (and potentially even the Ab+ dementia patients, if possible) and then assesses these relationships again? Age is such a murky variable in it's association with pathological change and it becomes very hard to assess these associations when age is such a large confound, as it is here between these groups, particularly considering the CU Ab- are so much younger

Authors' response: We agree that age is an important confound to consider. The CU Ab-group was so much younger because we kept our group of “young controls” that include young

and middle-aged adults. In confirmatory analyses, we now restricted the CU Ab- group to individuals age 50 and older, to match with the group of non-demented Ab+. The overall rate of change is the same in this group of older CU Ab- individuals. See Supplementary Fig. 1 (copied below) where we show the tau-PET data in CU Ab- restricted to those above 50, CU Ab+ and MCI Ab+ separately to be more explicit on the raw data. We refer to it in the legend of Figure 1 (page 8): “The group of Aβ-negative controls spans both middle-aged and older individuals. In Supplementary Fig. 1, we show individuals above 50 years old, as well as dividing the non-demented group into CU and MCI. “

We should mention that the group of CU Ab- is only showed as a reference and was not included in any analyses, so the composition of this group has minimal effect in this manuscript.

Supplementary Fig. 1. Mean spatial distribution of cross-sectional tau-PET [¹⁸F]RO948 SUVR and longitudinal rate of change in individual diagnosis groups

(A) Surface renderings of average baseline tau-PET SUVR in Aβ-negative controls above 50 years old, Aβ-positive CU and Aβ-positive MCI in the 200 parcels from the Schaefer 200-ROI atlas (B) Surface renderings of yearly tau-PET SUVR rate of change derived as the slope from linear mixed-effect models in the same participants group as in (A). CU=Cognitively unimpaired; MCI=Mild cognitive impairment

11. The mPACC is so much lower in the AB+ non-demented group that I really suggest to me that the MCI group is having a huge impact on this grouping. Is it at all possible to split these groups up or to just examine the impact of CU AB+ alone? That is, is this really just the MCIs telling us something (i.e. once cognitive impairment is already a factor)? I suspect that it is, and I feel that this is an important story to acknowledge.

Authors’ response: In line with response to point #4, we now repeated all main analyses splitting the group of non-demented into CU Ab+ and MCI Ab+ alone. Specifically related to cognitive decline, we see that in CU Ab+ alone, all analyses related to mPACC (now called cognitive composite score as per Reviewer 3’s suggestion) hold in this subgroup. In MCI Ab+,

all analyses hold with MMSE. These results have been added as Supplementary Fig. 11 and are mentioned in the text, see answer to point #4 for all details.

We agree that it is important to highlight that the main results hold already at the preclinical stage of AD and allude to this in the Discussion: *“All findings in non-demented group also even held when analyzing CU and MCI groups separately (Supplementary Fig. 6, 8 and 11)”* (page 18) and *“As such, A β -related increases in soluble p-tau may be a key initial step in the A β cascade that determines the accumulation of aggregated tau pathology, and thereby leading to faster cognitive decline in early AD, and this cascade of events was even found in the preclinical stage of the disease”* (page 19)

12. Given the association between Ab-PET and CSF p-tau, at least at the ‘earlier’ levels of Ab burden, did the authors consider how there might be shared variance explaining these ‘separate’ relationships with changes in tau-PET burden. That is, these analyses are non-independent and trying to examine the main effects in separate analyses might not be enough to tell us about the independent impacts of Ab-PET and CSF p-tau? Perhaps considerations of PCA or partial least squares (where a factor analysis of Ab and tau to extract shared and separate sources of variance) might help us to get at this question a bit better? My concern is particularly apparent for the analyses where both Ab and p-tau are included in the model as main effects. The loss of significance of one variable is potentially an artifact of warped estimates due to multicollinearity.

Authors’ response: That is indeed an important point. We ensured that our analyses are not affected by high multicollinearity between Ab-PET and CSF p-tau by using Variable Inflation Factors (VIF). Calculating VIF between p-tau and Ab SUVR across the 200 brain regions, we found limited effect of multicollinearity. VIFs ranged between 1.04 and 1.34 across the 200 regions, with an average of 1.16, representing low levels of multicollinearity.

As such, we believe that the analyses are suited to examine the independent impacts of Ab and CSF p-tau.

We added a sentence about this check for multicollinearity in the Methods section on Statistical analysis: *“We also ensured that there was no multicollinearity between A β -PET SUVR and CSF p-tau₂₁₇, with variance inflation factors ranging between 1.04 and 1.34.”* (page 28)

13. Did the authors ever examine CSF tau₂₁₇/Ab₄₂ ratio instead of just CSF₂₁₇? I am curious as to whether this might provide stronger associations or not.

Authors’ response: We had not considered it initially, and repeated the main analyses with this ratio, see figure below. We should note that this ratio is very strongly related to p-tau₂₁₇ (R=0.92, p<0.001) and less to Ab₄₂ (R=-0.55, p<0.001). It is also strongly related to global Ab PET SUVR (R=0.77, p<0.001). Overall, we see strong effects of this CSF ratio on subsequent tau aggregation rates in the non-demented stage of the disease, that also survives adjusting for baseline tau aggregates. However, since this ratio makes it harder to tease apart the effects of tau and amyloid separately, we have not included it in the manuscript.

14. Many of the Ab-PET ROIs are also non-independent either - that is, they are highly associated with one another and determining whether one region is truly more associated with changes in tau becomes quite difficult. Did the authors consider similar approaches to examining shared variance across regions?

Authors' response: We agree that many of the Ab-PET ROIs are strongly related to one another, with Ab PET binding already showing a widespread pattern of deposition early in the AD continuum. However, given that our analyses are focused on the local Ab- and tau-PET associations, and less on the shared variance across Ab SUVR across regions, we did specifically look at the inter-dependence of Ab-PET.

Still, in line with Reviewer 3 point #1, we corroborated the main results using the commonly used global Ab-PET SUVR instead of regional Ab, and the results were very similar. See Supplementary Fig. 5A, copied below.

Supplementary Fig. 5. CSF p-tau217 and A β associations using different measures of A β with regional tau-PET [18 F]RO948 rate of change in A β -positive non-demented participants

Similar analyses as done on Fig. 2, using different amyloid measure instead of regional A β -PET. The amyloid measure is global A β -PET SUVR in **A** and CSF A β 42/40 ratio in **B**. In both panels, the left two columns show standardized beta coefficient in regions where A β and CSF p-tau respectively relates to regional tau-PET rate of change, adjusting for age and sex. Two middle columns show standardized beta coefficient where A β and CSF p-tau217 are associated to regional tau-PET rate of change when including both biomarkers in the same model, adjusting for age and sex (tau PET rate of change \sim A β + CSF p-tau217 + age + sex). Right two columns show the same depiction as in middle columns when additionally controlling for regional baseline tau-PET SUVR. All regions shown on the brain are significant at p<0.05 after FDR-correction.

15. Can the authors report the correlation between p-tau217 and baseline tau-PET?

Authors' response: We have added the ROI-wise correlations between CSF p-tau217 and baseline tau-PET as Supplementary Fig. 4 (copied below) to complement Figure 2. Overall, the regions with the highest correlation between CSF p-tau and baseline tau-PET are also in line with the ones showing the highest with longitudinal tau-PET.

We also added the following sentence referring to these results (page 9): “*a similar effect of A β being no longer significant after accounting for soluble p-tau was found on baseline tau aggregates (Supplementary Fig. 4).*”

Supplementary Fig. 4. Regional associations with baseline tau-PET [^{18}F]RO948 SUVR in A β -positive non-demented participants

(A) Standardized beta coefficient of local A β -PET in regions where regional A β -PET flutemetamol SUVR (left column) relates to baseline tau-PET, adjusting for age and sex. Right column was derived from a similar model, but using CSF p-tau217 as predictor instead of A β -PET (B) Standardized beta coefficient where local A β -PET (left column) and CSF p-tau217 (right column) is associated to baseline tau-PET when including both biomarkers in the same model, adjusting for age and sex (tau PET ~ regional A β -PET + CSF p-tau217 + age + sex)

16. If one swapped out p-tau217 for baseline tau-PET in figure 3C, would there be more or less explained variance (i.e. R²)?

Authors' response: Replacing p-tau217 for baseline tau-PET (here we chose regions approximating a temporal meta-ROI), the explained variance would be similar. We found an association of std beta=-0.43 (p<0.001), R²=0.26 for baseline tau-PET and connectivity-based tau aggregation, compared to std beta =-0.44 (p<0.001), R²=0.23 for CSF p-tau217 in Fig. 3C.

Importantly, the association with CSF p-tau217 also remains when further adjusting for baseline tau-PET (beta value=-0.23 (p=0.04), R²=0.28), which we believe is the most relevant aspect for the current study. We mention this new finding in the Results section (page 12): “*This association also survives adjustment for baseline tau-PET (β =-0.23, p =0.04).*”

17. While there is an association found in Figures 2A and 2B, the findings are quite weak and involve a lot of heterogeneity that I think is worth discussing.

Authors' response: We believe the reviewer might have meant Figure 3A and B and we agree that the heterogeneity is worth discussing (Figure 3B specifically). Part of this heterogeneity is due to having both CU and MCI Ab+ in this group of non-demented participants; the individuals showing the highest negative correlations between tau-PET rate of change and connectivity are the MCI patients (we have now color-coded Figure 3B according to diagnosis). Further, we show that the overall association between connectivity to epicenters at the individual level and CSF p-tau217 holds in Ab+ CU or MCI alone, further supporting our main results. These results in CU and MCI separately have been added as Supplementary Fig. 8 (copied in point #4).

We added the following explanation in the Results section: “*The same associations were found in CU and MCI participants alone (Supplementary Fig. 8), suggesting that this effect is not only due to the MCI participants who tend to show greater association between connectivity to epicenter and tau-PET rate of change (more negative β -values on Fig. 3B).*” (page 12)

18. In Figures 4A-D could the authors color the scatterplot according to diagnostic/Ab group?

Authors' response: Figure 4 has been updated so that the scatterplots are color-coded according to diagnostic (CU Ab+ or MCI Ab+).

19. Could the authors examine whether the tau-PET Q1 rate of change mediation model (Fig 4G) performs any differently relative to the general metric of tau-PET rate of change (i.e. removing the connectivity component)? Or any of the other quartiles? I just want to ascertain how well these models perform relative more comparable mediation variables that are likely to explain the association between CSF p-tau and mPACC change.

Authors' response: That is indeed an important consideration. We repeated these same models using the average tau-PET rate of change in regions approximating the temporal meta-ROI commonly used (Jack et al, *Brain*, 2017; Ossenkoppele et al, *JAMA*, 2018). The mediation analysis yielded the same results ($c'-c=0.23$), when using tau-PET rate of change in the temporal ROI instead of Q1. We added these analyses as Supplementary Fig. 10 (copied below) and referred to it in the results section (page 14): “*all results were also validated using a commonly used temporal meta-ROI (see Supplementary Fig. 10)*”.

Supplementary Fig. 10. Cognitive decline analyses in Aβ-positive non-demented participants using tau-PET change in the temporal meta-ROI as mediator

Same analyses as in Fig. 4, when using tau-PET rate of change in the temporal meta-ROI (shown in yellow on the brains). Results are the same as when using tau-PET rate of change in Q1 (Fig. 4).

(A-B) Scatter plots of associations relevant to subsequent mediation analyses, beta coefficients from linear regressions adjusting for age, sex and education are reported. (A) Association between tau-PET rate of change in the temporal meta-ROI and rate of change on the cognitive composite score. (B) Association between CSF p-tau217 and tau-PET rate of change in the temporal meta-ROI. (C) Mediation analysis showing a full mediation of tau-PET rate of change in the temporal meta-ROI (58% proportion mediated) on CSF p-tau217 and cognitive decline. Repeating the same analyses using MMSE rate of change as the cognitive outcome yielded similar results ($c'-c = -0.30$ [95% CI -0.43, -0.16], $p < 0.001$).

20. If the authors remove the MCI patients, do the connectivity and cognitive findings hold up (i.e. entirely CU cohort)? The inclusion of the MCI participants will introduce extra variance to play with in these models, which is good, but creates a much easier path to finding significance when really one might be simply tapping into those who are already well on their way to a clinical diagnosis of dementia (i.e. very far from preclinical AD or the 'early stages of AD').

Authors' response: As stated earlier in the response to points #4 and 11, almost all analyses, including the connectivity-based findings (Fig. 3) and cognitive decline (Fig. 4) results were present in the CU and MCI Ab+ participants analyzed in separate groups. The only exception is the mediation of CSF p-tau217 and cognitive decline using the beta-value from connectivity-based tau-PET rate of change as the mediator (Fig. 4F), which was found only across all non-demented participants (CU and MCI together). The mediation models using two measures of tau-PET rate of change (in Q1 or the temporal meta-ROI) as mediators between CSF p-tau and cognitive decline in CU and MCI yielded the same results as from the whole group of non-demented participants. See our response to point #4 for all supplementary figures and text added to support this point.

We believe this is an important addition that strengthened our proposed model of tau pathology progression, as stated in the Discussion: *"All findings in non-demented group also even held when analyzing CU and MCI groups separately (Supplementary Fig. 6, 8 and 11)" (page 18) and "As such, A β -related increases in soluble p-tau may be a key initial step in the A β cascade that determines the accumulation of aggregated tau pathology, and thereby leading to faster cognitive decline in early AD, and this cascade of events was even found in the preclinical stage of the disease" (page 19)*

21. The authors mention soluble levels of CSF Ab42/40 in their dementia section but it is not analyzed or mentioned in the previous sections - why is this?

Authors' response: We apologize for the confusion regarding the use of CSF Ab in the dementia group only. Initially, we introduced the CSF Ab42/40 ratio only in the dementia section because the AD patients do not undergo Ab PET and only had CSF measures of Ab. In response to Reviewer 1 point #1, we repeated the analyses based on Ab PET using the CSF Ab42/40 ratio in the non-demented group (see Supplementary Fig. 5 copied in response to point #14), which now improves consistency in our analyses in the pre-dementia and dementia stage.

We also clarified in the Methods that the reason of using only CSF A β in the dementia stage is because AD dementia patients do not undergo A β PET: *"AD patients did not undergo A β -PET, and thus for this group only, analyses were restricted to CSF A β 42/40" (page 25).*

22. The authors mention ADNI early in their results section but I do not see mention of it again

in the results? I may be missing something but it would be worth focusing some sentences on validation for each analysis throughout the section.

Authors' response: It is true that initially we kept the ADNI results mostly in Supplementary material. We now added a few sentences at the end of each Results section to summarize the key findings from ADNI, which reads as follows:

(page 6) In ADNI (Supplementary Table 1), the spatial pattern of the main regions with highest tau-PET binding at baseline and the highest rate of change in tau aggregates per year mirrored the patterns found in BioFINDER-2 (Supplementary Fig. 3). However, the magnitude of accumulation of tau aggregates was almost twice as small in ADNI compared to BioFINDER-2.

(page 9) In ADNI, as in BioFINDER-2, soluble p-tau, measured with CSF p-tau181, was the main factor related to regional accumulation of tau aggregates over time (Supplementary Fig. 7). Analyses were conducted using two A β measures: the global centiloid (CL) score since two different PET A β tracers are used in ADNI, and CSF A β 42. In both cases, soluble p-tau remained significant when accounting for A β and baseline tau-PET SUVR in temporoparietal regions, although the regional pattern was more restricted than in BioFINDER-2 (Supplementary Fig. 7).

(page 12-13) Results were also validated in the A β -positive non-demented participants from ADNI (Supplementary Fig. 9). The overall connectivity-based association with tau-PET rate of change (β -value) was related to the levels of soluble p-tau (standardized coefficient = -0.24, $p=0.01$), there was an interaction between soluble p-tau and connectivity-based tau aggregates accumulation by quartiles ($p=0.01$).

(page 15) Based on effect sizes determined in BioFINDER-2, sample sizes between 75 and 82 participants would have been needed in order to assess effects of soluble p-tau on cognitive decline in ADNI. Since only 28 participants had longitudinal cognitive assessments, this set of analyses was not conducted in ADNI.

23. The authors mention early in their discussion that ADNI “fully” replicated their results, but there should be caveats here to remind the reader that ADNI is a smaller sample size and that only certain analyses were able to be validated in this cohort due to power issues. Further, this replication really isn't mentioned at all in the results and needs to be explicitly described. The authors allude to the shortcomings in the limitations section, but needs to be amplified more throughout the text.

Authors' response: We removed the word “fully” from the sentence in the discussion and as stated above, we now more clearly describe the results and shortcomings throughout the Results section.

Reviewer #3:

Pichet Binette et al. report longitudinal data from the BIOFINDER2 cohort. The PET and fMRI data are based on a brain parcellation atlas. fMRI connectivity is analysed using correlation analysis. CSF 217phosphotau is a predictor of connectivity-dependent increase in tau PET signal and has a stronger effect than regional amyloid PET values.

This multimodal longitudinal dataset is very rich in content and relatively unique. The analyses are thorough and well developed. The main findings are based on state-of-the-art statistics (incl. mediation modeling). The findings are novel and of clinical and fundamental relevance and are presented with sufficient detail. My main concern relates to the way in which amyloid load is parameterized throughout the paper.

Major comments

1. The comparison between 217phosphotau as predictor versus regional Abeta is not entirely symmetrical. There are as many regional A β values as regions examined (n = 200), compared to a single CSF parameter. For each of the analyses, the authors should verify whether results remain if A β load is expressed as a single measure based on a composite region. A composite amyloid PET value may be more robust than a set of 200 regional values. This could be the typical composite region used for amyloid imaging or, in the asymptomatic stage, precuneus levels. It is conceivable that A β measures after parcellation into 200 regions yields a substantial portion of regional levels below the cut-off and hence reflects mostly noise. That would inevitably lead to poor results for the A β predictor compared to phosphotau. Furthermore, the regional tau accumulation is correlated with the amyloid load in the same region. The underlying assumption that amyloid accumulation and tau accumulation colocalize, is unlikely.

Authors' response: Thank you for the positive and constructive comments on our manuscript. The point regarding the parametrization of amyloid is well taken and we have addressed this in the revised version. We have replicated our initial results using and global score of A β PET as suggested by the reviewer (and also the CSF A β 42/40 ratio as suggested by Reviewer 1) instead of the regional A β PET measures. Overall, regardless of the chosen measure of A β , we consistently see the importance of CSF p-tau217 in subsequent tau-PET rate of change. These results have been as Supplementary Fig. 5, copied below, as we added the following text in the Results section: "*These results were also consistent across different measures of A β , i.e. if using global A β load assessed by PET or the CSF A β 42/40 ratio instead of regional A β aggregates in regression models (Supplementary Fig. 5).*" (page 9)

Supplementary Fig. 5. CSF p-tau217 and A β associations using different measures of A β with regional tau-PET [18 F]RO948 rate of change in A β -positive non-demented participants

Similar analyses as done on Fig. 2, using different amyloid measure instead of regional A β -PET. The amyloid measure is global A β -PET SUVR in **A** and CSF A β 42/40 ratio in **B**. In both panels, the left two columns show standardized beta coefficient in regions where A β and CSF p-tau respectively relates to regional tau-PET rate of change, adjusting for age and sex. Two middle columns show standardized beta coefficient where A β and CSF p-tau217 are associated to regional tau-PET rate of change when including both biomarkers in the same model, adjusting for age and sex (tau PET rate of change \sim A β + CSF p-tau217 + age + sex). Right two columns show the same depiction as in middle columns when additionally controlling for regional baseline tau-PET SUVR. All regions shown on the brain are significant at p<0.05 after FDR-correction.

2. Data from cognitively normal participants and MCI are pooled. There is a strong interest in the earliest stages of AD, when participants are still cognitively normal. Hence it would be critical to verify which results remain when only the cognitively normal participants are examined. Given the relatively slow rate of change of tau load in the cognitively normal stage, it is conceivable that the relations described are no longer significant. That would have a profound repercussion on the interpretation as that would change the paper from a study of the earliest pathophysiological changes in AD to a study of changes occurring during the clinical stages of AD. This of course would remain of interest but the neurobiological interpretation would become different. The designation ‘non-demented participants’ may be technically correct but from a clinical perspective, however the distinction between a cognitively normal, community-recruited participant versus a memory clinic recruited patient with mild cognitive deficits is crucial. The grouping as ‘non-demented’ is meant to increase sensitivity by including symptomatic cases who are known to have higher biomarker levels that change more rapidly but is scientifically contentious. Pooling MCI and cognitively normal makes it difficult for the reader to determine whether the findings pertain in the first instance to the asymptomatic AD stage or to the MCI stage.

Authors’ response: We agree that this is an important consideration, and one that was also raised by Reviewer 2 in several points. We have thus repeated all analyses by splitting CU and MCI. All main results (strong effect of CSF p-tau on tau-PET rate of change, association between connectivity-based tau-PET change and CSF p-tau levels, mediation effect of tau-PET rate of change on CSF p-tau and cognitive decline) were found in each separate group, which we believe strengthens the neurobiological interpretation put forward.

All these results have been added as Supplementary Fig. 6, 8 and 11, copied below and followed by the corresponding text added in the Results section. Please see also our answer to Reviewer 2 point #4 for complementary information.

Supplementary Fig. 6. Regional A β -PET and CSF p-tau217 associations with regional tau-PET [18 F]RO948 rate of change in A β -positive CU and MCI separately

Same analyses as reported in Fig. 2 in the full sample of non-demented A β -positive participants, when splitting the group into CU only (**A**) and MCI participants only (**B**). In both panels, the left two columns show standardized beta coefficient in regions where regional A β -PET and CSF p-tau217 respectively relates to regional tau-PET rate of change, adjusting for age and sex. Two middle columns show standardized beta coefficient where regional A β -PET and CSF p-tau217 are associated to regional tau-PET rate of change when including both biomarkers in the same model, adjusting for age and sex (tau PET rate of change \sim regional A β -PET + CSF p-tau217 + age + sex). Right two columns show the same depiction as in middle columns when additionally controlling for regional baseline tau-PET SUVR. All regions shown on the brain are significant at $p < 0.05$ after FDR-correction.

Supplementary Fig. 8. Individualized connectivity-based associations of tau-PET rate of and CSF p-tau217 in CU and MCI Aβ-positive participants separately

Same analyses as on Fig. 3A-C in the full sample of non-demented Aβ-positive participants, done here in CU Aβ-positive only (**A**) and MCI Aβ-positive (**B**). All analyses were significant in CU and MCI separately. In both panels, left scatter plot depicts group-level analysis showing how connectivity to the tau epicenters relates to tau-PET rate of change across the whole brain. Each dot represents a brain region. Regions more strongly functionally connected to the epicenters have greater rate of tau-PET accumulation. Repeating the same approach depicted at the individual level, we could generate for each participant a β -value from the correlation between tau-PET rate of change and connectivity-based distance to epicenters across all brain regions, as shown on the box plot in the middle. Right scatter plots show the association between CSF p-tau217 and the β -values of epicenter connectivity to tau-PET rate of change. Each dot represents an individual. The expected negative association suggests that higher CSF p-tau217 is associated with the overall pattern of tau-PET change in more functionally connected regions to epicenters.

A CU Aβ+**B MCI Aβ+****Supplementary Fig. 11. Cognitive decline analyses in Aβ-positive CU and MCI separately**

Same mediation analyses as in Fig. 4 in the full sample of non-demented Aβ-positive participants, done here in CU Aβ-positive only (**A**) and MCI Aβ-positive (**B**). Main analyses were done using tau-PET rate of change in Q1, as shown on the graphs, but also hold when instead using tau-PET rate of change in the temporal meta-ROI. In CU, results were found with rate of change on the cognitive composite score or MMSE. In MCI, results were only found with MMSE rate of change.

Text added in the Results section of the main manuscript:

Further, investigating the same relations in the CU and MCI groups separately, the strong effect of soluble p-tau on subsequent tau aggregation rate, above the effect of Aβ, was clearly found in both groups (Supplementary Fig. 6). When further adjusting for local baseline tau aggregates, soluble also remained a significant predictor in temporo-parietal regions in each group (Supplementary Fig. 6). (page 9)

The same associations were found in CU and MCI participants alone (Supplementary Fig. 8), suggesting that this effect is not only due to the MCI participants who tend to show greater association between connectivity to epicenter and tau-PET rate of change (more negative β-values on Fig 3B). (page 12)

The same analyses were also repeated splitting the non-demented group into Aβ-positive CU and MCI separately. We observe a full mediation effect of tau-PET rate of change, either in Q1 or in the temporal meta-ROI between soluble p-tau and cognitive decline on the cognitive composite score in CU ($c' - c = -0.21 [-0.40, -0.06], p = 0.002$), and on decline on MMSE in MCI ($c' - c = -0.24 [-0.46, -0.04], p = 0.01$), see Supplementary Fig. 11 for detailed statistics. The only analysis that we could not replicate in individual groups was the mediating effect of the β-value of connectivity-based tau-PET change (Fig. 4F), which was also of smaller magnitude in the full group of non-demented participants. (page 14)

3. The authors push the interpretation of their data in terms of the underlying neurobiological processes and the pathogenesis of Alzheimer disease. This is laudable and exciting and ideally should be possible but with the current strength of evidence for these biomarkers, the authors could be more cautious about the interpretation of biomarkers in terms of underlying neurobiological processes and early AD pathogenesis. Among their four biomarkers, only amyloid PET is neuropathologically validated by an end-of-life validation data. Amyloid PET

mostly detects amyloid stage 4 and 5 with high accuracy and amyloid stage 3 with moderate accuracy. Hence, the amyloid measure does not detect the earliest stages as one may think based on the neurobiological interpretation the authors assign to their in vivo measures. For the other biomarkers, neuropathological validation is missing. For the tau PET tracer, it is plausible that it detects only Braak stage 4 and 5 reliably by analogy with flortaucipir but we (at least the reviewer) do not know for sure. For soluble 217phosphotau there are no end-of-life data that validate this CSF measure as a reliable indicator of a specific neurobiological process in the brain. In clinical practice we assume that it is AD specific if a stringent cut-off is used, but the meaning of CSF phosphotau as a continuous measure of the level of a specific compound in the brain is not proven strictly speaking. This is a matter of opinion, there is no strong objective evidence to the best of my knowledge and it would be hard to collect such evidence. If there is strong evidence of a continuous relationship between CSF 217phosphotau and brain levels of soluble hyperphosphorylated tau, please cite it in the paper. For the fMRI connectomics the evidence and technical validation in terms of neuroanatomical brain connectivity is also limited to the best of our knowledge, while such evidence would be essential for an interpretation in terms of transsynaptic spread. Again, please cite the relevant evidence demonstrating the validity of the brain fMRI interregional correlation as a measure of mono- or multisynaptic anatomical connectivity in the paper.

Authors' response: Both for tau-PET and soluble p-tau217 or 181, there is good evidence between these in vivo biomarkers and end-of-life data from neuropathology. With flortaucipir, many studies have shown correspondence between flortaucipir and postmortem Braak stages (Fleisher et al, *JAMA Neurology*, 2021), or regional flortaucipir and p-tau measured in corresponding tissue postmortem (Pontecorvo et al, *EJNMMI Research*, 2022).

With soluble p-tau, our group has also shown strong association between antemortem p-tau217 (from plasma) and tau density tangles from neuropathology in two different studies, both with Spearman correlations above 0.65 (Pamqvist et al, *JAMA*, 2020: Figure 2; Wennstrom et al, *Acta Neuropathol Commun*, 2022: Table 4). Also, in both cases the results suggested a continuous association between soluble p-tau and tangles, supporting our choice of analyzing CSF p-tau as a continuous measure. Other groups have also showed associations between soluble p-tau181 and neuropathological tau postmortem (Lantero-Rodriguez et al, *EMBO Mol Med*, 2020; Grothe et al, *Neurology*, 2021; Smirnov et al, *Acta Neuropathol*, 2022). We added the following sentence in the Introduction to clarify this point (page 3): *“Importantly, levels of soluble p-tau have been shown to correlate with neuropathological levels of insoluble fibrillar tau¹⁴⁻¹⁶”*

Regarding functional connectivity, we did not mean to suggest that from human imaging data functional connectivity reflects direct synaptic connectivity per se, but rather that measures of brain connectivity have been linked to patterns of tau deposition and accumulation, making it an interesting approach to try to further understand tau accumulation. We explain it in the Introduction (page 3): *“We and others reported that patient-level tau accumulation is related to the connectivity patterns of regions where tau aggregates emerge first (i.e, tau epicenters)²⁷⁻²⁹.”* In addition, we have added a sentence to the limitation section of the Discussion, highlighting that fMRI-based connectivity is an indirect measure of neuronal connectivity and potentially also captures multi-synaptic indirect connections: *“Lastly, we acknowledge that functional connectivity is an indirect measure of brain activity, but still has been shown to be related to mono- and polysynaptic pathways (Grandjean et al, *J Neuroscience*, 2017).”*

Minor comments

4. Like other groups, the authors describe an acceleration of tau PET tracer signal increase with disease stage based on linear change in the asymptomatic, the MCI and the early dementia stage. Across the entire cohort is the increase best captured by an exponential model? If the rate constant of the exponential model is used as outcome measure rather than the linear change, is there still an effect of the same baseline variables? Or is the effect of the baseline variables entirely attributable to where the case is situated on the exponential curve?

Authors' response: While there is likely an exponential relation between tau-PET deposition and disease stage cross-sectionally (as nicely shown for instance in Doré et al, *Eur J Nucl Med Mol Imaging*, 2019), we do not believe that tau PET tracer rate of change follows an exponential increase. For instance, the associations between baseline tau-PET and tau-PET rate of change in a temporal meta-ROI show more of a linear association, as seen on the left scatterplot below. Similar associations are also seen if we compare baseline tau SUVR in individualized epicenters and tau-PET rate of change in the different quartiles, with Q1 being shown on the right scatterplot below

Previous studies investigating the association between baseline tau-PET and tau-PET rate of change also suggest linear associations between the two measures (Smith et al, *Brain*, 2020: Figure 2A; Pontecorvo et al, *Brain*, 2019: Figure 2A; Cho et al, *J Nucl Med*, 2019).

We thought this minor comment was perhaps beyond the scope of the current manuscript and have not added it to the main manuscript.

Association between tau-PET baseline SUVR and rate of change across the whole sample

Temporal meta-ROI

Individualized ROIs

5. If baseline tau PET load is included as one of the predictors in the mediation model (Fig 4E-G), does the effect of phosphotau remain or does baseline tau PET load as predictor wipe out all the other effects?

Authors' response: We appreciate the reviewers concern, since indeed baseline tau-PET is an important marker of future cognitive decline (Ossenkoppele et al, *JAMA Neurol*, 2020; Hanseeuw et al, *JAMA Neurol*, 2019). Baseline tau-PET SUVR is strongly correlated with soluble p-tau ($r=0.78$), and very much so to tau-PET rate of change ($r=0.92$). However, including baseline tau-PET as an additional predictor in the mediation model to predict cognitive decline will result in multicollinear predictors which makes it difficult to interpret individual regression weights of either p-tau or tau-PET. Supporting this, neither soluble p-tau, baseline tau-PET, or tau-PET rate of change become significant when all measures are included in the mediation model, suggesting that the specific variance that explains future cognitive changes is mostly shared among both p-tau and baseline tau-PET. More importantly, we would

like to note that our analyses were mainly motivated to test a biological cascade model in which A β -related elevation of soluble p-tau precede the aggregation of tau and might relate to the subsequent aggregation rate of tau, regardless of the amount of tau aggregates already in the brain. Therefore, we believe that our mediation models should be restricted to include baseline p-tau as a main predictor.

In addition, we would like to emphasize that the purpose of this analysis was not to define the best predictor of cognitive change in AD (in which case baseline tau-PET would indeed be a key predictor), but to assess a cascade model in which Ab drives soluble p-tau elevations, which in turn results in higher tau aggregation rates and therefore cognitive decline.

6. The term ‘modified PACC’ is misleading as the set of cognitive tests differs too much from the PACC. I realize this term has been used by this group in previous study but it should be re-named in a more objective manner rather than suggesting a high similarity with the PACC.

Authors’ response: We changed the term ‘modified PACC’ for ‘Cognitive composite score’ throughout the manuscript. In the first instance that the score is mentioned and in the Methods, we specified that we used “*a cognitive composite score analogous to the PACC5*”. (page 4 and 23)

7. Overall the average interval of follow-up remains relatively modest (1.5 yrs), especially for the asymptomatic phase that extends over many years, with a clinical phase that can be 10 years or longer.

Authors’ response: It is indeed a limitation of our study, which we now address in the Discussion (see text below).

We would also like to highlight that in this revised version we updated our dataset with the latest tau-PET and cognitive assessment, which raised our average follow-up to 2 years for tau-PET, and 3 years for cognitive assessment (PET scans are done every 2 years unlike cognitive visit). While this is still short for the preclinical/prodromal phase, it is in line with the longest tau-PET follow-ups in currently published studies (e.g. Sanchez et al, *STM*, 2021, Jack et al, *Brain*, 2020).

“First, although we are expanding on previous studies that were limited by shorter follow-up or smaller sample sizes⁴⁷⁻⁴⁹, a longer follow-up time of tau-PET would increase the rate of tau aggregates accumulation in earlier stages of AD, which spans many years.” (page 20)

8. The fMRI connectivity matrix is based on correlations between time series. This is used as a measure of transsynaptic spread. Logically, partial correlations would be more suited as correlations can be direct or indirect and for transsynaptic spread the distinction between mono- and multi-synaptic connectivity is relevant. Second, correlations between timeseries can be heavily influenced by Euclidean distance. Please explain how you deal with this potential confound.

Authors’ response: Following the reviewers’ comment, we generated the functional connectivity matrix template using partial correlations between preprocessed resting-state fMRI timeseries of each ROI. The connectivity matrix using partial correlations was highly correlated with the original matrix (R=0.8), suggesting that using partial or full correlations yields similar connectivity matrices. We reconducted the main connectivity-based analyses using the partial correlation matrix, and results were the same, as shown in the Table below.

The correlation between the individualized beta-values derived with the standard or partial correlations were also very highly correlated, $R=0.96$.

Regarding Euclidean distance, we would like to highlight that the Euclidean distance between ROIs was included as a covariate to calculate the beta-coefficient between regional tau-PET change and connectivity to the tau epicenters at the group level in Fig. 3A, and it was not a significant predictor.

Further, we also reconducted the main analyses generating the individualized beta value from the region-wise association between connectivity to the epicenters and tau-PET rate of change when adjusting for Euclidean distance between ROIs, and the main results are also the same, as shown in the Table below. The correlation between the individualized beta-values derived with or without adjusting for Euclidean distance were also highly correlated, $R=0.86$.

	Connectivity from « standard » correlations (Main manuscript)	Connectivity from partial correlations	Connectivity from « standard » correlations + adjusting for Euclidean distance
Individual level – Fig. 3C Beta-value from connectivity and CSF p-tau217	-0.44	-0.34	-0.35
Individual level – Fig. 4C Beta-value from connectivity and cognitive decline	0.34	0.32	0.40

To not further complicate an already complex set of analyses, we did not include these confirmatory analyses in the manuscript but restrict it to the rebuttal. Please note that the rebuttal will be made available online in case the manuscript is accepted for publication, so these additional confirmatory results will be available for the interested reader.

Still, we added a sentence in the Methods section pointing that the results were replicated when using partial correlations or adjusting for Euclidean distance:

“Note that repeating this same approach when adjusting for Euclidean distance between ROIs, or when using a functional connectivity matrix based on partial correlations as template, results were unchanged.” (page 28)

9. P 26: the authors compare models with different sets of predictors. Please explain how you compare the difference in fit between models and when you decide that the fit differs significantly.

Authors’ response: Those models refer to the results presented on Figure 2, and our objective was to investigate how the strength of predictors vary when controlling for different covariates in the models, but not to compare model fits per se. Specifically, we were interested in the effect of amyloid and CSF p-tau217 when both are predictors of change in tau-PET, and when further adjusting for baseline tau-PET in the models. As such, we rather focus on the individual predictors with the rate of tau accumulation rather than the overall model fit.

10. Please add the correlation matrix between the baseline variables. The paper is presented as separating out effects of different baseline variables but if these are highly correlated, the distinctions may be less clear-cut. It is also relevant for the mediation model.

Authors' response: We have added the associations of Ab-PET and CSF p-tau217 with baseline tau-PET across the 200 brain regions in line with point #15 of Reviewer 2, and the results have been added as Supplementary Fig. 4. We also added the correlation between various baseline and longitudinal measures in Supplementary Table 2 copied below, which show moderate associations overall. In the Methods section, we added that “*Correlations between baseline and longitudinal variables are also reported for descriptive purposes in Supplementary Table 2.*” (page 29)

Supplementary Fig. 4. Regional associations with baseline tau-PET [¹⁸F]RO948 SUVR in Aβ-positive non-demented participants

(A) Standardized beta coefficient of local Aβ-PET in regions where regional Aβ-PET flutemetamol SUVR (left column) relates to baseline tau-PET, adjusting for age and sex. Right column was derived from a similar model, but using CSF p-tau217 as predictor instead of Aβ-PET (B) Standardized beta coefficient where local Aβ-PET (left column) and CSF p-tau217 (right column) is associated to baseline tau-PET when including both biomarkers in the same model, adjusting for age and sex (tau PET ~ regional Aβ-PET + CSF p-tau217 + age + sex)

Supplementary Table 2. Bivariate correlations in Ab-positive non-demented participants

	R	p-value
Baseline cognitive composite score vs. CSF p-tau217	-0.23	0.01
Baseline MMSE vs. CSF p-tau217	-0.05	0.56
Baseline cognitive composite score vs. baseline tau-PET temporal ROI	-0.29	<0.001
Baseline MMSE vs. baseline tau-PET temporal ROI	-0.21	0.02
Tau-PET temporal ROI: Baseline vs. rate of change	0.94	<0.001
Baseline tau-PET temporal ROI vs. CSF p-tau217	0.65	<0.001

11. Fig 4 panel A-D: please clarify whether the association is based on Pearson or Spearman correlation. Given the distribution of the data, this certainly should be based on Spearman correlations.

Authors' response: The associations in Figure 4 are based on Pearson correlations, but all associations remain with Spearman correlations. In panels A-D, difference in correlation coefficients are minor between the two methods (see Table below for comparisons of bivariate correlations). We clarified in the legend of the figure that “*beta coefficients from linear regressions are reported*” (page 15).

	Panel A	Panel B	Panel C	Panel D
	CSF p-tau217 vs. cognitive change	Tau-PET change vs. cognitive change	Beta-connectivity vs. cognitive change	CSF p-tau217 vs. Tau-PET change
Pearson correlation	-0.41	-0.43	0.34	0.56
Spearman correlation	-0.38	-0.34	0.35	0.51

Very minor comment

12. When change in tau PET is modelled, the authors use ‘time in years’ as independent variable. The time resolution should be clarified. Ideally one would use scan date and model change over time in that way I guess.

Authors' response: The reviewer is correct that we have used the time difference in scan date (in years) to compute tau-PET change rates. We clarified this aspect in the Methods section “Regional measures and rate of change of PET SUVR”: “*To calculate the rate of change in tau-PET over time, linear mixed effect models with random slope and intercept were fitted for each brain region, with tau-PET SUVR as the dependent variable and time in years from the baseline scan date as the independent variable.*” (page 26)

13. To define tau PET centers the authors used a mixed Gaussian model based on the data they have ‘since the right-most distribution likely reflects abnormal tau PET signal’ (p 24-25). Using an independent dataset to define thresholds of tau pathology per region would be a good alternative.

Authors' response: The method we used did not rely on defining thresholds. For each participant, we used the Gaussian-mixture modeling probability from each region as a continuous measure. This allowed to identify the regions with the highest probability of being “high tau” at baseline at the individual level, and we considered those regions as epicenters. We clarified this on page 27: “*Since this right-most distribution likely reflects abnormal tau-PET signal, the GMM probability represents the probabilistic measure of tau positivity, without the need to use a priori thresholds.*”

REVIEWER COMMENTS

Reviewer #1 (Remarks to the Author):

I believe the authors addressed my comments and the manuscript can be accepted in its current form.

Reviewer #2 (Remarks to the Author):

I very much appreciate the authors taking the time to really explicitly state what their innovative addition to the field is with this paper, and this has settled a lot of things in my mind. Unfortunately, one phrase within this argument has exposed another question for me that I would appreciate a response to (below). And a few other comments follow.

The authors make the comment that "A β -related increased soluble levels of p-tau are a key driver in the accumulation of tau aggregates", but I'm not entirely convinced that these analyses are showing this. The first set of analyses tests the fixed effects of Ab-PET (for instance) on tau slopes and then the second tests the included covariance of CSF p-tau. This set of analyses do not suggest that Ab-related increases are a key driver. They suggest that after covarying for CSF p-tau, there is no explained variance left from Ab. Then, the authors present a mediation, but do not display the results of the direct and indirect effects. They simply state that across all voxels, the average proportion of mediation by p-tau on the relationship between Ab and tau change. The issue I have here is that there is not a clear representation of (on average) what the association is between Ab signal and p-tau signal (pathway a) so that it is unclear whether this mediation is holding up. I assume that there is, but the authors could perhaps make this more explicit. These are questions that arise in my mind as I am going through the results again.

On the final line of page 5, the authors use the phrase "gradually extending" to describe cross-sectional data, but this is misleading.

I very much appreciate seeing the analyses stratified by diagnostic group and I agree with the authors that this is also a novel addition to the manuscript.

I would respectfully push back on the authors that the cerebellar grey reference region is ideal for longitudinal tau-PET. For instance, this excellent work by the Mayo group (<https://alz-journals.onlinelibrary.wiley.com/doi/full/10.1002/alz.054086>) has examined many different approaches for examining tau PET SUVR change over time (not with RO948, mind you - this is with FTP), and it shows in Figure 2 that there is less reliable performance when using cerebellar crus reference region. Our own lab (unpublished) has had similar results. Some work from Hanseeuw and colleagues tends to use WM (doi:10.1001/jamaneurol.2019.1424). As such, I would not assume that inferior cerebellum will be the ideal candidate for FTP. I am less clear on RO948, but appreciate the authors addressing this issue further. They argue that they assessed this issue for their paper in 2020 but there are two differences here: the previous paper did not examine RO948 and there was far less follow up. Now with much more observational follow-up, and a different tracer, does this remain the case?

I am very glad to hear that PVC was not used for longitudinal tau-PET! Please see previous Mayo reference in agreement with authors.

Reviewer #3 (Remarks to the Author):

The authors have addressed my comments satisfactorily.

Reviewer #1:

I believe the authors addressed my comments and the manuscript can be accepted in its current form.

Authors' response: We thank the reviewer for judging our revised manuscript acceptable for publication.

Reviewer #2:

I very much appreciate the authors taking the time to really explicitly state what their innovative addition to the field is with this paper. and this has settled a lot of things in my mind. Unfortunately, one phrase within this argument has exposed another question for me that I would appreciate a response to (below). And a few other comments follow.

Authors' response: We are glad that the reviewer found our explanation helpful and that the novelty of our findings to the field now comes across more clearly. We are thankful for the opportunity to clarify the few remaining points below.

The authors make the comment that “A β -related increased soluble levels of p-tau are a key driver in the accumulation of tau aggregates”. but I'm not entirely convinced that these analyses are showing this. The first set of analyses tests the fixed effects of Ab-PET (for instance) on tau slopes and then the second tests the included covariance of CSF p-tau. This set of analyses do not suggest that Ab-related increases are a key driver. They suggest that after covarying for CSF p-tau, there is no explained variance left from Ab. Then, the authors present a mediation, but do not display the results of the direct and indirect effects. They simply state that across all voxels, the average proportion of mediation by p-tau on the relationship between Ab and tau change. The issue I have here is that there is not a clear representation of (on average) what the association is between Ab signal and p-tau signal (pathway a) so that it is unclear whether this mediation is holding up. I assume that there is, but the authors could perhaps make this more explicit. These are questions that arise in my mind as I am going through the results again.

Authors' response: We agree that we should have been more explicit in reporting the full results of the mediation analyses. We now provide summary statistics from the mediations in Supplementary Fig. 8, copied below. We averaged the coefficients and p-values across all regions where CSF p-tau₂₁₇ was significantly mediating Ab-PET and tau-PET rate of change. It is now clearer that all pathways meet the requirement to perform mediation analyses.

Supplementary Fig. 8. Mediating effect of CSF p-tau217 on Aβ-PET and regional tau-PET [¹⁸F]RO948 rate of change in Aβ-positive non-demented participants

Summary statistics of the mediation from Fig. 2d. The coefficients and p-values reported are from the average across regions where CSF p-tau217 has a significant mediating effect.

On the final line of page 5, the authors use the phrase “gradually extending” to describe cross-sectional data, but this is misleading.

Authors’ response: We removed this misleading phrasing. The sentence now reads “... the baseline distribution of insoluble tau aggregates assessed via tau-PET recapitulated the AD-typical deposition in the medial and lateral temporal lobes in the controls and non-demented participants, and into lateral and medial parietal and lateral occipital regions at symptomatic AD stages”

I very much appreciate seeing the analyses stratified by diagnostic group and I agree with the authors that this is also a novel addition to the manuscript.

Authors’ response: We thank the reviewer for suggesting these analyses in the first round of revisions.

I would respectfully push back on the authors that the cerebellar grey reference region is ideal for longitudinal tau-PET. For instance, this excellent work by the Mayo group (<https://alz-journals.onlinelibrary.wiley.com/doi/full/10.1002/alz.054086>) has examined many different approaches for examining tau PET SUVR change over time (not with RO948, mind you - this is with FTP), and it shows in Figure 2 that there is less reliable performance when using cerebellar crus reference region. Our own lab (unpublished) has had similar results. Some work from Hanseeuw and colleagues tends to use WM (doi:10.1001/jamaneurol.2019.1424). As such, I would not assume that inferior cerebellum will be the ideal candidate for FTP. I am less clear on RO948, but appreciate the authors addressing this issue further. They argue that they assessed this issue for their paper in 2020 but there are two differences here: the previous paper did not examine RO948 and there was far less follow up. Now with much more observational follow-up, and a different tracer, does this remain the case?

Authors’ response: We agree with the reviewer that inferior cerebellar grey matter might not be the best reference region for longitudinal analyses, and we did not mean to imply that it was the case. We are, however, cautious with using white matter reference regions, despite the important work by the Mayo group cited by the reviewer. Specifically, our concern is based on previous work showing that the tau-PET signal in white matter depends on white matter integrity (i.e. the presence of white matter intensities). A previous study using Flortaucipir showed tracer binding affinity to myelin, and that Flortaucipir retention is lower

in white matter hyperintensities compared to normal appearing white matter (Moscoso et al., *EJNMMI*, 2021). Even further, longitudinal Flortaucipir signal decrease was stronger in white matter hyperintensities than in normal appearing white matter, suggesting that longitudinal tau-PET quantification using a white matter reference region can be confounded by the presence of white matter hyperintensities and vascular co-pathology that is highly prevalent in aging populations. Therefore, we refrained from using an eroded white matter reference region but prefer to keep our main analyses using a cerebellar grey reference.

Nevertheless, we tried to address the reviewers concern by systematically investigating the effect of different reference regions on tau-PET change rates, including the eroded white matter reference region cited by the reviewer. Specifically, we considered regions of interest that approximate Braak stages (I-II, III-IV and V-VI) based on the paper by Cho et al. *Annals of Neurology*. 2016. We calculated rate of change in SUVR/year from RO948 scans referenced to either (1) inferior cerebellar grey matter, (2) whole cerebellum (grey and white matter combined) and (3) eroded white matter derived as in the abstract cited by the reviewer. We report below the correlations on rate of change based across these three SUVR options obtained with different reference regions and the average rate of tau-PET change in different diagnostic groups.

Overall, using inferior cerebellum cortex and whole cerebellum as reference regions yielded almost identical rates of change, as found with FTP (Young et al, *Neuroimage*, 2021). The correlations between the cerebellar reference regions and the eroded white matter were also very high (all correlations ranging between 0.88 and 0.93) (Table 1 below). Looking at rates of change by diagnostic groups (Table 2), we see slightly higher rates of change using inferior cerebellum cortex or whole cerebellum compared to eroded white matter, but the latter still yields comparable rates of change.

We should also mention a very recent publication investigating Flortaucipir rates of change across different cohorts in preclinical AD (Insel et al, *Brain*, 2022), in which there was virtually no difference if using the cerebellum gray matter or eroded white matter as reference regions. The only differences found were in the controls with very low amyloid (a group that we do not perform any major analysis in the current manuscript), where potentially a slightly higher rates of change could be detected using eroded white matter as the reference region (0.005 vs 0.006 SUVR/year in the inferior temporal lobe).

One argument against longitudinal analyses using inferior cerebellar cortex as reference region could also be that the region is rather small, and therefore a small change in the registration of PET to MRI could affect the SUV in that region. Using the whole cerebellum should mitigate that, and the very high correlations between both methods – and with white matter reference region – speak against that we have a systematic error that is introduced by poor image registration. We (specifically authors APB and RS) also perform visual quality control of every PET scan (and its registration to MRI) to mitigate problems with misregistration.

Taken together, considering the latest literature (mostly based on Flortaucipir) and our comparisons done in this response (using RO948), we believe that the inferior cerebellum reference region is adequate to measure reliable tau-PET SUVR rate of change over time, although we agree that this is an important methodological question to continue exploring in future studies.

Table 1. Correlation coefficients between rates of change generated from SUVR images using different reference regions

	Inferior cerebellum cortex vs. Whole cerebellum	Inferior cerebellum cortex vs. eroded white matter	Whole cerebellum vs. eroded white matter
I-II region	0.98	0.90	0.93
III-IV region	0.99	0.91	0.93
V-VI region	0.98	0.88	0.92

Table 2. Rates of change generated from SUVR images using different reference regions by diagnostic groups

	Rate of change - inferior cerebellum reference region	Rate of change - whole cerebellum reference region	Rate of change - eroded white matter reference region
I-II region			
CU A β negative	0.0093	0.0083	0.0088
CU A β positive	0.0202	0.0184	0.0187
MCI A β positive	0.0281	0.0265	0.0264
AD A β positive	0.0399	0.0391	0.0366
III-IV region			
CU A β negative	0.0073	0.0053	0.0048
CU A β positive	0.0230	0.0235	0.0214
MCI A β positive	0.0492	0.0509	0.0482
AD A β positive	0.1097	0.1047	0.0911
V-VI region			
CU A β negative	0.0042	0.0033	0.0032
CU A β positive	0.0107	0.0101	0.0087
MCI A β positive	0.0272	0.0255	0.0217
AD A β positive	0.0770	0.0743	0.0546

All values represent SUVR/year

I am very glad to hear that PVC was not used for longitudinal tau-PET! Please see previous Mayo reference in agreement with authors.

Authors' response: We agree with the reviewer.

Reviewer #3:

The authors have addressed my comments satisfactorily.

Authors' response: We thank the reviewer for judging our revised manuscript acceptable for publication.

REVIEWERS' COMMENTS

Reviewer #2 (Remarks to the Author):

Very happy with these in depth responses to my second round of review. I sincerely appreciate the time taken and the consideration that the authors took to reply to my comments. Thank you!